# Universal generalization guarantees for Wasserstein distributionally robust models

**Tam Le**   **Jérôme Malick**
Univ. Grenoble Alpes, CNRS, Grenoble INP, LJK
Grenoble, 38000, France

## Abstract

Distributionally robust optimization has emerged as an attractive way to train robust machine learning models, capturing data uncertainty and distribution shifts. Recent statistical analyses have proved that generalization guarantees of robust models based on the Wasserstein distance have generalization guarantees that do not suffer from the curse of dimensionality. However, these results are either approximate, obtained in specific cases, or based on assumptions difficult to verify in practice. In contrast, we establish exact generalization guarantees that cover a wide range of cases, with arbitrary transport costs and parametric loss functions, including deep learning objectives with nonsmooth activations. We complete our analysis with an excess bound on the robust objective and an extension to Wasserstein robust models with entropic regularizations.

## 1 Introduction

### 1.1 Wasserstein robustness: models and generalization

Machine learning models are challenged in practice by many obstacles, such as biases in data, adversarial attacks, or data shifts between training and deployment. Towards more resilient and reliable models, distributionally robust optimization has emerged as an attractive paradigm, where training no longer relies on minimizing the empirical risk but rather on an optimization problem that takes into account potential perturbations in the data distribution; see e.g., the review articles Kuhn et al. (2019); Blanchet et al. (2021a).

More specifically, the approach consists in minimizing the worst-risk among all distributions in a neighborhood of the empirical data distribution. A natural way (Mohajerin Esfahani & Kuhn, 2018) to define such a neighborhood is to use the optimal transport distance, called the Wasserstein distance (Cédric & Villani, 2009). Between two suitable distributions $Q$ and $Q'$ on a sample space $\Xi$, we may define the optimal transport cost among all coupling $\pi$ on $\Xi \times \Xi$ having $Q$ and $Q'$ as marginals:

$$W_c(Q, Q') = \inf_{\substack{\pi \in \mathcal{P}(\Xi \times \Xi) \\ [\pi]_1 = Q, [\pi]_2 = Q'}} \mathbb{E}_{(\xi, \zeta) \sim \pi}[c(\xi, \zeta)], \tag{1}$$

where $c \colon \Xi \times \Xi \to \mathbb{R}$ is a non-negative cost function. When $c$ is the power $p \geq 1$ of a distance on $\Xi$, this corresponds to the $p$-Wasserstein distance. For a class of loss functions $\mathcal{F}$, the Wasserstein distributionally robust counterpart of the standard empirical risk minimization (ERM) then writes

$$\min_{f \in \mathcal{F}} \quad \sup_{Q \in \mathcal{P}(\Xi), W_c(\widehat{P}_n, Q) \leq \rho} \mathbb{E}_{\xi \sim Q}[f(\xi)], \tag{2}$$

for a chosen radius $\rho$ of the Wasserstein ball centered at the empirical data distribution, denoted $\widehat{P}_n$. This procedure is often referred to as Wasserstein Distributionally Robust Optimization (WDRO). In the degenerate case $\rho = 0$, we have $Q = \widehat{P}_n$ and (2) boils down to ERM. If $\rho > 0$, the training captures data uncertainty and provides more resilient learning models; see e.g. the discussions and illustrations in Shafieezadeh-Abadeh et al. (2015); Sinha et al. (2018); Zhao & Guan (2018); Kwon et al. (2020); Li et al. (2020); Taskesen et al. (2021); Gao et al. (2022); Arrigo et al. (2022); Belbasi et al. (2023).

To support theoretically the modeling versatility and the practical success of these robust models, some statistical guarantees have been proposed in the literature. For a population distribution $P$, i.i.d. samples $\xi_1, \ldots, \xi_n$ drawn from $P$, and the associated empirical distribution $\widehat{P}_n := \frac{1}{n} \sum_{i=1}^{n} \delta_{\xi_i}$, the best concentration results for the Wasserstein distance (Fournier & Guillin, 2015) gives that if the radius $\rho$ is large enough, then the Wasserstein ball around $\widehat{P}_n$ contains the true distribution $P$ with high probability, which in turn gives directly (see Mohajerin Esfahani & Kuhn (2018)) a generalization bound of the form

$$\sup_{Q \in \mathcal{P}(\Xi), W_c(\widehat{P}_n, Q) \leq \rho} \mathbb{E}_{\xi \sim Q}[f(\xi)] \geq \mathbb{E}_{\xi \sim P}[f(\xi)]. \tag{3}$$

This bound is exact in the sense that it introduces no approximation term between the true risk and the robust risk, unlike standard generalization bounds of ERM (Vapnik (1999); Bartlett & Mendelson (2006)). This property (3) is specific to WDRO and highlights its potential to give more resilient models: the left-hand-side, which is the quantity that we compute from data and optimize by training, provides a control on the right-hand-side which is the idealistic population risk.

In order to obtain such an attractive guarantee, Mohajerin Esfahani & Kuhn (2018) needs to take a large radius $\rho$. Indeed, the direct application of concentration results of Fournier & Guillin (2015) requires a radius scaling as $O(1/n^{\frac{1}{d}})$, where $d$ is the data dimension. Due to the exponential dependence, in high dimension, this value is almost constant with respect to $n$, hence suggesting that the exact bound (3) can hold only for conservative values of $\rho$.

Recent works have proposed various statistical guarantees for WDRO by establishing generalization bounds that do not suffer from the above curse of the dimension; we further discuss them in the related work section in section 1.3. These results generally feature a radius $\rho$ scaling as $O(1/\sqrt{n})$, which is the standard rate in ERM generalization bounds. Yet, no existing result on robust models precisely retrieve the original exact bound (3) with the $1/\sqrt{n}$ rate, in a general setting.

## 1.2 Contributions and outline

In this paper, we establish exact generalization guarantees of the form (3) under general assumptions that cover many machine learning situations. Our results apply to any kind of data lying in a metric space (e.g. classification and regression tasks with mixed features) and general classes of continuous loss functions (e.g. from standard regression tasks to deep learning models) as long as standard compactness conditions are satisfied. For instance, our results cover nonsmooth objectives that are particularly present in deep learning with the use of $\mathrm{ReLU}$ activation function, max-pooling operator, or optimization layers.

To avoid using concentration results of Fournier & Guillin (2015) involving a radius scaling as $O(1/n^{\frac{1}{d}})$, we develop a novel optimization-based proof, directly tackling the nonsmoothness of the robust objective function (2) with tools from variational analysis (Clarke, 1990; Rockafellar & Wets, 1998; Aliprantis & Border, 2006). We thus obtain general results with $\rho$ scaling as $O(1/\sqrt{n})$, capturing all possible nonsmoothness and coinciding with previous study for robust linear models (Shafieezadeh-Abadeh et al., 2019).

Moreover, our approach is systematic enough to (i) provide estimates of the excess errors quantifying by how much the robust objective may exceed the true risk, and (ii) extend to the recent versions of Wasserstein/Sinkhorn distributionally robust problems that involve (double) regularizations (Azizian et al., 2023b; Wang et al., 2023). We thus complete the only existing analysis of regularized WDRO (Azizian et al., 2023a) by obtaining generalization results for double regularization (Azizian et al., 2023b) when dealing with arbitrary costs and nonsmooth objectives.

The paper is structured as follows. First, Section 2 introduces and illustrates the setting of this work. Then Section 3 presents and discusses the main results: the generalization guarantees (Theorem 3.1 and Theorem 3.2), the excess risk bounds (Proposition 3.1 and Proposition 3.3) and the specific case of linear models (Section 3.2). This section ends with Section 3.4 discussing the limitations of our study and potential extensions. Finally, Section 4 highlights our proof techniques, combining classical concentration lemma and advanced nonsmooth analysis.

## 1.3 RELATED WORK

The majority of papers studying generalization bounds of Wasserstein distributionally robust models establish *approximate* generalization bounds. These approximate bounds introduce vanishing terms depending on $n$ and $\rho$ which embody the bias of WDRO. One of the first papers on such approximate bounds is Lee & Raginsky (2018), and important results in this direction include Blanchet et al. (2021b); Blanchet & Shapiro (2023) about asymptotical results for smooth losses, and Chen & Paschalidis (2018) about non-asymptotically results for linear models and for smooth loss functions. Let us also mention Yang & Gao (2022) which deals with 0-1 loss, and Gao (2022) which focuses on Wasserstein-1 uncertainty and the connection with Lipschitz norm regularization. In this paper, we rather focus of *exact* bounds of the form of (3), which are out of reach of ERM-based models, and thus capture the essence of WDRO.

The literature about exact bounds is scarcer than the one about approximate bounds and significantly different in terms of proof techniques. Let us mention Shafieezadeh-Abadeh et al. (2019) which establishes exact guarantees for linear regression models, and Gao (2022) which proposes tighter results for linear regression and Wasserstein-1 uncertainty. The closest work to our paper is Azizian et al. (2023a), which initiates a general study on exact bounds. There, the authors establish generalization results similar to ours, namely: exact bounds (3) in a regime where $\rho > O(1/\sqrt{n})$. In sharp contrast with our work, the results from Azizian et al. (2023a) rely on restrictive assumptions to overcome the nonsmoothness of the robust objective: the squared norm for the cost $c$, a Gaussian reference distribution, additional growth conditions, and abstract compactness conditions. We will further compare the setting and the results, in Section 2 and Section 3 and in the supplemental. In our work, we directly deal with nonsmoothness, thanks to tools from nonsmooth analysis, and thus we are able to alleviate extra assumptions and capture nonsmooth losses.

The only other work regarding nonsmooth objectives is An & Gao (2021) which derives results on piece-wise smooth losses, at the price of abstract approximating constants. We underline that none of the existing results properly covers nonsmooth losses, in particular deep learning objectives with nonsmooth activations.

Finally, let us mention that there exist many works studying generalization guarantees for other distributionally robust models, involving different uncertainty quantification. For instance, Zeng & Lam (2022) studies nonparametric families and divergence-based ambiguity, and Bennouna et al. (2023) considers deep learning models with ambiguity sets that combine KL divergence and adversarial corruptions. Though duality is always an important tool, we face in our framework to the specific difficulty of dealing with Wasserstein distances, so that the technicalities as well as the results of our paper are essentially different and disjoint from these works.

## 1.4 NOTATIONS

Throughout the paper $(\Xi, d)$ is a metric space, where $d$ is a distance, $\mathcal{F}$ is a family of loss functions $f : \Xi \to \mathbb{R}$ and $c : \Xi \times \Xi \to \mathbb{R}$ is a cost function.

**On probability spaces.** We denote the space of probability measures on $\Xi$ by $\mathcal{P}(\Xi)$. For all $\pi \in \mathcal{P}(\Xi \times \Xi)$, $i \in \{1, 2\}$, we denote the $i^{\text{th}}$ marginal of $\pi$ by $[\pi]_i$. We denote the Dirac mass at $\xi \in \Xi$ by $\delta_\xi$. Given a measurable function $g : \Xi \to \mathbb{R}$, we denote the expectation of $g$ with respect to $Q \in \mathcal{P}(\Xi)$ by $\mathbb{E}_{\xi \sim Q}[g(\xi)]$ and we may also use the shorthand $\mathbb{E}_Q[g]$.

**On function spaces.** For a function $f : \Xi \to \mathbb{R}$, we denote the uniform norm by $\|f\|_\infty := \sup_{\xi \in \Xi} |f(\xi)|$. By extension, we use the notation $\|\mathcal{F}\|_\infty := \sup_{f \in \mathcal{F}} \|f\|_\infty$. Whenever well-defined, we denote the set of maximizers of $f$ on $\Xi$ by $\arg\max_\Xi f := \{\zeta \in \Xi : f(\zeta) = \max_{\xi \in \Xi} f(\xi)\}$.

We say $f$ is *Lipschitz* with constant $L > 0$ if for all $\xi, \zeta \in \Xi$, $|f(\xi) - f(\zeta)| \leq L d(\xi, \zeta)$. For a function $\phi$, we denote $\partial_\lambda^+ \phi$ the right-sided derivative with respect to $\lambda \in \mathbb{R}$, and $\partial_\lambda \phi$ its derivative, whenever well-defined.

## 2 Assumptions and examples

In this section, we present the general framework and illustrate it by standard examples. We make the following assumptions on the sample space $\Xi$, the space of loss functions $\mathcal{F}$ and the cost $c$.

**Assumption 2.1.**

1. $(\Xi, d)$ *is compact.*

2. $c$ *is jointly continuous with respect to $d$, non-negative, and $c(\xi, \zeta) = 0$ if and only if $\xi = \zeta$.*

3. *Every $f \in \mathcal{F}$ is continuous and $(\mathcal{F}, \|\cdot\|_\infty)$ is compact. Furthermore, if $N(t, \mathcal{X}, \|\cdot\|_\infty)$ is the $t$-packing number of $\mathcal{F}$[1], then Dudley's entropy of $\mathcal{F}$,*

$$\mathcal{I}_\mathcal{F} := \int_0^\infty \sqrt{\log N(t, \mathcal{X}, \|\cdot\|_\infty)} dt,$$

*is finite.*

Our setting encompasses many machine learning scenarios, with parametric models, general loss functions, and general transport costs as illustrated in the two paragraphs below. In Section 3.4, we will also come back on these assumptions to discuss their reach and limitations.

Before this, let us underline that this set of assumptions is quite general and allows us to conduct a unified study and to relieve several restrictions found in previous works. In particular, we mention that the results of the closest work Azizian et al. (2023a) require convexity of $\Xi$, differentiability of losses $f \in \mathcal{F}$, restriction to the squared Euclidean distance, together with a strong[2] structural property on $\mathcal{F}$ (Azizian et al., 2023a, Assumption 5), aimed at overcoming the nonsmoothness of the WDRO objective. In our sketch of proof in Section 4, we explain how we deal directly with nonsmoothness.

**Parametric models and loss functions.** Our setting covers a wide range of machine learning models. Consider a parametric family $\mathcal{F} = \{f(\theta, \cdot) : \theta \in \Theta\}$, where the parameter space $\Theta \subset \mathbb{R}^p$ is compact and the loss function $f : \Theta \times \Xi \to \mathbb{R}$ is jointly Lipschitz continuous. Since $\Xi$ is compact, such a family is compact regarding $\|\cdot\|_\infty$ and $\mathcal{I}_\mathcal{F}$ is finite, proportional to $\sqrt{p}$. This situation covers regression models, k-means clustering, and neural networks. For example: least-squares regression

$$f(\theta, (x, y)) = (\langle \theta, x \rangle - y)^2, \quad \Xi \subset \mathbb{R}^m \times \mathbb{R},$$

logistic regression

$$f(\theta, (x, y)) = \log\left(1 + e^{-y\langle\theta, x\rangle}\right), \quad \Xi \subset \mathbb{R}^m \times \{-1, 1\},$$

and support vector machines with hinge loss

$$f(\theta, (x, y)) = \max\{0, 1 - y\langle\theta, x\rangle\}, \quad \Xi \subset \mathbb{R}^m \times \{-1, 1\}.$$

Note that the latter is not differentiable, due to the max term. The k-means model also introduces a non-differentiable loss function:

$$f(\theta, x) = \min_{i \in \{1, \ldots, K\}} \|\theta_i - x\|_2^2, \quad \Theta \subset \mathbb{R}^{K \times m}, \Xi \subset \mathbb{R}^m.$$

Finally, most deep learning models fall in our setting. Indeed, they involve loss functions of the form

$$f(\theta, (x, y)) = \ell(h(\theta, x), y),$$

where $\ell$ is a dissimilarity measure and $h$ is a parameterized prediction function, built as a composition of affine transformations (which are the parameters to train) with activation functions (see e.g. Krizhevsky et al. (2012); LeCun et al. (2015); Redmon et al. (2016)). Our setting is general enough to encompass all continuous activation functions, even non-differentiable ones (as ReLU $= \max(0, \cdot)$) as well as other nonsmooth elementary blocks (as max-pooling (He et al., 2016), sorting procedures (Sander et al., 2023), and optimization layers (Amos & Kolter, 2017)). As already underlined in introduction, these examples involving non-differentiable terms are not covered by existing results.

---

[1]The maximal number of functions in $\mathcal{F}$ that are at least at a distance $t$ from each other.

[2]We show in Proposition F.5 in the appendix that the compactness assumptions (Azizian et al., 2023a, Assumption 5) hide strong conditions on the maximizers.

**Sample space and transport costs.** The choice of the transport cost $c$ depends on the nature of the data and of the potential data uncertainty. For instance, if the variables are continuous with $\Xi \subset \mathbb{R}^m$, we consider the distance $d = \| \cdot - \cdot \|_p$ induced by $\ell_p$-norm ($p \in [1, \infty]$) and the cost as a power ($q \in [1, \infty)$) of the distance

$$c(\xi, \xi') = \|\xi - \xi'\|_p^q.$$

If the variables are discrete with $\Xi \subset \{1, \ldots, J\}^m$, we consider the distance

$$d(\xi, \xi') = \sum_{i=1}^{m} \mathbb{1}_{\{\xi_i \neq \xi_i'\}}$$

and the cost as a power of this distance. If we deal with mixed data, i.e. they contain both continuous and discrete variables, a sum of the previous costs can be considered. In classification, for instance, with the samples composed of features $x \in \mathbb{R}^m$ and a target $y \in \{-1, 1\}$, we may take

$$c((x, y), (x', y')) = \|x - x'\|_p^q + \kappa \mathbb{1}_{\{y \neq y'\}}$$

for a chosen $\kappa > 0$. This cost is obviously continuous with respect to the natural distance

$$d((x, y), (x', y')) = \|x - x'\|_p + \mathbb{1}_{\{y \neq y'\}}.$$

This extends to mixed data with categorical, binary and continuous variables; see e.g. Belbasi et al. (2023).

## 3 MAIN RESULTS

### 3.1 WASSERSTEIN ROBUST MODELS

Our main result establishes a generalization bound (3) for Wasserstein distributionally robust optimization (WDRO). Given a distribution $Q \in \mathcal{P}(\Xi)$ and a loss $f \in \mathcal{F}$, the robust risk around $Q$ with radius $\rho > 0$ is defined as

$$R_{\rho,Q}(f) := \sup_{Q' \in \mathcal{P}(\Xi), W_c(Q,Q') \leq \rho} \mathbb{E}_{\xi \sim Q'}[f(\xi)]. \tag{4}$$

In particular, taking $Q = \widehat{P}_n$ and $Q = P$ in the above expression, we consider the empirical robust risk, $\widehat{R}_\rho(f)$, and the true robust risk, $R_\rho(f)$:

$$\widehat{R}_\rho(f) := R_{\rho,\widehat{P}_n}(f) \quad \text{and} \quad R_\rho(f) := R_{\rho,P}(f).$$

We also introduce the following constant, called the *critical radius* $\rho_{\text{crit}}$,

$$\rho_{\text{crit}} := \inf_{f \in \mathcal{F}} \mathbb{E}_{\xi \sim P} \left[ \min \left\{ c(\xi, \zeta) \ : \ \zeta \in \arg\max_\Xi f \right\} \right]. \tag{5}$$

Note that $\rho_{\text{crit}}$ is defined from the true distribution $P$, which makes it a deterministic quantity. In our results, we will make the further assumption that $\rho_{\text{crit}} > 0$, which excludes[3] losses that remain constant across all samples from the ground truth distribution $P$. This assumption reasonably aligns with practice and is also in line with the previous works An & Gao (2021); Azizian et al. (2023a). For instance, obtaining a predictor that precisely interpolates the ground truth distribution (leading to a loss equal to zero everywhere) is unrealistic. In this context, our main result then establishes the generalization bound when $n$ is large enough, for $\rho$ scaling with the standard $1/\sqrt{n}$ rate.

**Theorem 3.1** (Generalization guarantee for Wasserstein robust models). *If Assumption 2.1 holds and $\rho_{\text{crit}} > 0$, then there exists $\lambda_{\text{low}} > 0$ such that when $n > \frac{16(\alpha+\beta)^2}{\rho_{\text{crit}}^2}$ and $\rho > \frac{\alpha}{\sqrt{n}}$, we have with probability at least $1 - \delta$,*

$$\widehat{R}_\rho(f) \geq \mathbb{E}_{\xi \sim P}[f(\xi)] \qquad \text{for all } f \in \mathcal{F},$$

*where $\alpha$ and $\beta$ are the two constants*

$$\alpha = 48 \left( \|\mathcal{F}\|_\infty + \frac{1}{\lambda_{\text{low}}} \right) \left( \mathcal{I}_\mathcal{F} + \frac{2}{\lambda_{\text{low}}} \right) + \frac{2\|\mathcal{F}\|_\infty}{\lambda_{\text{low}}} \sqrt{2 \log \frac{2}{\delta}}, \qquad \beta = \frac{96 \mathcal{I}_\mathcal{F}}{\lambda_{\text{low}}} + \frac{4\|\mathcal{F}\|_\infty}{\lambda_{\text{low}}} \sqrt{2 \log \frac{4}{\delta}}.$$

---

[3] See Proposition F.3, in Appendix F.1 about the interpretation of the critical radius.

This result with $\rho$ scaling with the dimension-free $1/\sqrt{n}$ rate is similar to (Azizian et al., 2023a, Theorem 3.1), but guaranteed now in the wide setting of Assumption 2.1. We achieve this result through a novel proof technique that combines nonsmooth analysis rationale with classical concentration results; as depicted in Section 4.

The critical radius $\rho_{\text{crit}}$ can be interpreted as a degeneracy threshold of the robust problem; we discuss it below in Remark 3.1. The quantity $\lambda_{\text{low}}$ is a positive constant related to the geometry of the Wasserstein ambiguity set; we discuss it in Section 4.2. Interestingly, in the case of linear and logistic regressions, we can establish estimates of these two quantities; see Section 3.2.

We now extend the previous result to derive the following excess risk bound.

**Proposition 3.1** (Excess risk for Wasserstein robust models)**.** *Let $\alpha$ be given by Theorem 3.1. Under Assumption 2.1, if $\rho_{\text{crit}} > 0$, $n > \frac{16\alpha^2}{\rho_{\text{crit}}^2}$ and $\rho \leq \frac{\rho_{\text{crit}}}{4} - \frac{\alpha}{\sqrt{n}}$, then with probability at least $1 - \delta$,*

$$\widehat{R}_\rho(f) \leq R_{\rho + \frac{\alpha}{\sqrt{n}}}(f) \qquad \text{for all } f \in \mathcal{F}.$$

*In particular, if $c = d(\cdot, \cdot)^p$ with $p \in [1, \infty)$ and every $f \in \mathcal{F}$ is $\text{Lip}_{\mathcal{F}}$-Lipschitz, then*

$$\widehat{R}_\rho(f) \leq \mathbb{E}_{\xi \sim P}[f(\xi)] + \text{Lip}_{\mathcal{F}} \left( \rho + \frac{\alpha}{\sqrt{n}} \right)^{\frac{1}{p}}.$$

**Remark 3.1** (The critical radius as a degeneracy threshold)**.** *The critical radius $\rho_{\text{crit}}$ defined in (5) plays the role of a degeneracy threshold for the problem, as follows. We can show that if $\rho \geq \rho_{\text{crit}}$, there exists $f \in \mathcal{F}$ satisfying $R_\rho(f) = \max_{\xi \in \Xi} f(\xi)$. Furthermore, for $\rho \geq \rho_{\text{crit}} + \frac{\alpha}{\sqrt{n}}$, with high probability, there exists $f \in \mathcal{F}$ such that $\widehat{R}_\rho(f) = \max_{\xi \in \Xi} f(\xi)$. In other words, if the radius is chosen too high compared to $\rho_{\text{crit}}$, both generalization and excess bounds (Theorem 3.1 and Proposition 3.1) are vacuous. A proof of this result is found in Appendix, Proposition F.1.*

### 3.2 GENERALIZATION GUARANTEES OF WASSERSTEIN ROBUST LINEAR MODELS

We now illustrate how our generalization guarantees from Section 3.1 apply to linear models. In this part, we assume the support of $P$ to be contained in a ball of diameter $D$ centered at $0$.

We recover estimates similar to the ones from the study of linear models (Shafieezadeh-Abadeh et al., 2019), hence showing the tightness of our approach. We consider the setting from Shafieezadeh-Abadeh et al. (2019) where the parameter space is assumed to be bounded away from zero (Shafieezadeh-Abadeh et al., 2019, Assumption 4.5):

**Assumption 3.1** (Hypothesis domain)**.** $\mathcal{F} = \{ f(\theta, \cdot) : \theta \in \Theta \}$, *where $\Theta \subset \mathbb{R}^p$ is a compact subset and $c = \| \cdot - \cdot \|^2$. There exists $\omega > 0$ satisfying one of the following:*

1. *(Linear regression). $f(\theta, (x, y)) = (\langle \theta, x \rangle - y)^2$, $\inf_{\theta \in \Theta} \|(\theta, -1)\|^2 \geq \omega$ and $\omega \geq 1$.*

2. *(Logistic regression). $f(\theta, (x, y)) = \log(1 + e^{-y\langle \theta, x \rangle})$ and $\inf_{\theta \in \Theta} \|\theta\|^2 \geq \omega$.*

Under this assumption, we obtain estimates for the constants $\lambda_{\text{low}}$ and $\rho_{\text{crit}}$.

**Proposition 3.2** (Linear models dual bound and critical radius)**.** *Under Assumption 3.1, let $\Omega := \sup_{\theta \in \Theta} \|\theta\|^2$. Theorem 3.1 and Proposition 3.1 hold with $\rho_{\text{crit}} \geq D^2$ and*

1. *(Linear regression) $\lambda_{\text{low}} \geq \frac{\omega}{2}$ under Assumption 3.1.1.*

2. *(Logistic regression) $\lambda_{\text{low}} \geq \frac{\omega}{8\left(1 + e^{D\Omega}\right)}$ under Assumption 3.1.2.*

These specific results show that we retrieve the constants from Shafieezadeh-Abadeh et al. (2019), for normalized data in the case of logistic regression. In particular, our constant $\alpha$ is proportional to $1/\omega^2$ for the linear regression. Remark that the tails parameters of $f(\xi)$ and $\xi \sim P$ from Shafieezadeh-Abadeh et al. (2019) correspond in our case to $\|\mathcal{F}\|_\infty$ and $D$ respectively, and Dudley's constant is proportional to $\sqrt{p}$. In the case of linear regression, the dual lower bound is directly related to the parameter bound $\omega$ from Shafieezadeh-Abadeh et al. (2019). In more advanced settings (e.g. deep learning), the positivity of $\lambda_{\text{low}}$ can be seen as an implicit definition of the hypothesis bound $\omega$.

### 3.3 REGULARIZED WASSERSTEIN ROBUST MODELS

Part of the success of optimal transport in machine learning is the use of regularization, and specifically entropic regularization, opening the way to nice properties and efficient computational schemes (Cuturi, 2013; Peyré & Cuturi, 2019). Recall that the entropy-regularized optimal transport cost writes, for a reference coupling $\pi_0 \in \mathcal{P}(\Xi \times \Xi)$ as

$$W_c^\tau(P, Q) = \min_{\substack{\pi \in \mathcal{P}(\Xi \times \Xi) \\ [\pi]_1 = P, [\pi_2] = Q}} \{\mathbb{E}_\pi[c] + \tau D_{\mathrm{KL}}(\pi \| \pi_0)\} \tag{6}$$

where $D_{\mathrm{KL}}(\cdot \| \pi_0)$ is the Kullback-Leibler divergence w.r.t. $\pi_0$:

$$\mathrm{KL}(\pi \| \pi_0) = \begin{cases} \int_{\Xi \times \Xi} \log \frac{\mathrm{d}\pi}{\mathrm{d}\pi_0} \, \mathrm{d}\pi & \text{when } \pi \ll \pi_0 \\ \infty & \text{otherwise.} \end{cases} \tag{7}$$

Given the definition of $\mathrm{KL}(\cdot \| \pi_0)$, note that the minimum (6) is well-defined, attained at some coupling $\pi^{P,Q} \ll \pi_0$, see e.g. Peyré & Cuturi (2019). Regularization have been recently studied in the context of WDRO: Wang et al. (2023) introduces an entropic regularization in constraints for computational interests, Azizian et al. (2023a) considers an entropic regularization in the objective for generalization, and Azizian et al. (2023b) studies a general regularization in both constraints and objective.

Following the most general case (Azizian et al., 2023b) we consider the robust risk with double regularization

$$R_{\rho,Q}^{\tau,\epsilon}(f) := \sup_{\substack{\pi \in \mathcal{P}(\Xi \times \Xi), \, [\pi]_1 = Q \\ \mathbb{E}_\pi[c] + \tau \, \mathrm{KL}(\pi \| \pi_0) \leq \rho}} \{\mathbb{E}_{[\pi]_2}[f] - \epsilon D_{\mathrm{KL}}(\pi \| \pi_0)\}$$

with two parameters $\epsilon > 0$ and $\tau \geq 0$. We introduce the conditional moment[4] of $\pi_0$

$$m_c := \max_{\xi \in \Xi} \mathbb{E}_{\zeta \sim \pi_0(\cdot | \xi)}[c(\xi, \zeta)],$$

and the *regularized critical radius*

$$\rho_{\mathrm{crit}}^{\tau,\epsilon} := \inf_{f \in \mathcal{F}} \mathbb{E}_{\xi \sim P} \left[ \mathbb{E}_{\zeta \sim \pi_0^{f/\epsilon}(\cdot | \xi)} \left[ \frac{\tau}{\epsilon} f(\zeta) + c(\xi, \zeta) \right] - \tau \log \mathbb{E}_{\zeta \sim \pi_0(\cdot | \xi)} \left[ e^{\frac{f(\zeta)}{\epsilon}} \right] \right], \tag{8}$$

where $\mathrm{d}\pi_0^{f/\epsilon}(\cdot | \xi) \propto e^{f/\epsilon} \mathrm{d}\pi_0(\cdot | \xi)$. In this setting, the generalization guarantee states as follows.

**Theorem 3.2** (Generalization for double regularization). *Under Assumption 2.1, there exist $\alpha^{\tau,\epsilon} > 0$ and $\beta^{\tau,\epsilon} > 0$ depending on $\mathcal{F}, \Xi, c, \epsilon, \tau$ and $\delta$, such that if $\rho_{\mathrm{crit}}^{\tau,\epsilon} > 4m_c$, when $n > \frac{16(\alpha^{\tau,\epsilon} + \beta^{\tau,\epsilon})^2}{(\rho_{\mathrm{crit}}^{\tau,\epsilon} - 4m_c)^2}$ and $\rho > \max\left\{m_c, \frac{\alpha^{\tau,\epsilon}}{\sqrt{n}}\right\}$, we have with probability at least $1 - \delta$, for all $Q \in \mathcal{P}(\Xi)$ such that $W_c^\tau(P, Q) \leq \rho$,*

$$\widehat{R}_\rho^{\tau,\epsilon}(f) \geq \mathbb{E}_{\zeta \sim Q}[f(\zeta)] - \epsilon D_{\mathrm{KL}}(\pi^{P,Q} \| \pi_0) \qquad \text{for all } f \in \mathcal{F},$$

*where $\pi^{P,Q}$ is the optimal coupling in (6).[5]*

This result is similar to the one of Theorem 3.1 and is also similar to the only other generalization result existing for regularized WDRO (Azizian et al., 2023a). Let us explicit below the main differences.

Unlike Wasserstein robust models (Theorem 3.1), regularization leads to an *inexact* generalization guarantee, where the regularized empirical robust risk bounds a proxy for the true risk $\mathbb{E}_P[f]$. This is in line with the regularization in optimal transport that induces a bias in the Wasserstein metric, preventing $W_c^\tau(P, P)$ from being null. In particular, given an arbitrary $\tau > 0$, $W_c^\tau(P, P)$ may not be lower than $\rho$.

The coefficients $\alpha^{\tau,\epsilon}$ and $\beta^{\tau,\epsilon}$ exhibit similar relations with $\lambda_{\mathrm{low}}^{\tau,\epsilon}, \|\mathcal{F}\|_\infty$, and $\mathcal{I}_\mathcal{F}$ to their counterparts $\alpha$ and $\beta$ from Theorem 3.1. Their complete expressions can be found in the extended version

---

[4]E.g., if $c(\xi, \zeta) = \frac{1}{2}\|\xi - \zeta\|^2$ and $\pi_0(\cdot | \xi)$ is a truncated Gaussian $\pi_0(\cdot | \xi) \propto e^{-\frac{\|\cdot - \xi\|^2}{2\sigma^2}} \mathbb{1}_\Xi$, we have $m_c \propto \sigma^2$.
[5]Due to the definition of the Kullback-Leibler divergence (7), we necessarily have $\pi^{P,Q} \ll \pi_0$.

of Theorem 3.2 (in Appendix, Theorem E.2). In particular, the expression of $\alpha^{\tau,\epsilon}$ suggests $m_c$, $\epsilon$, $\tau$ and $\rho$ should be of comparable order. Compared to the standard setting, we have an estimate of the lower bound $\lambda_{\text{low}}^{\tau,\epsilon}$ (Lemma D.2) showing dependence on the loss family: $\lambda_{\text{low}}^{\tau,\epsilon} = O\big(e^{-\frac{\|\mathcal{F}\|_\infty}{\epsilon}}\big)$.

Compared to Azizian et al. (2023a), we underline that our result covers the double regularization case. Moreover, it is valid for an arbitrary $\pi_0$ whereas the one from Azizian et al. (2023a) relies on the specific form of $\pi_0$ involving a Gaussian term. Our result is thus more flexible, allowing freedom in the choice of $\pi_0$.

As for the standard case (Proposition 3.1), we obtain an excess risk bound. The main difference in this setting is that we lose the explicit control of the true risk. This is mainly due to the inexactness brought by regularization.

**Proposition 3.3** (Excess risk for doubly regularized robust models). *Let $\alpha^{\tau,\epsilon}$ be given by Theorem 3.2. Under Assumption 2.1, if $\rho_{\text{crit}}^{\tau,\epsilon} > 4m_c$, $n > \frac{16\alpha^{\tau,\epsilon 2}}{(\rho_{\text{crit}}^{\tau,\epsilon} - 4m_c)^2}$ and $m_c < \rho \leq \frac{\rho_{\text{crit}}^{\tau,\epsilon}}{4} - \frac{\alpha^{\tau,\epsilon}}{\sqrt{n}}$, then with probability at least $1 - \delta$,*

$$\widehat{R}_\rho^{\tau,\epsilon}(f) \leq R_{\rho + \frac{\alpha^{\tau,\epsilon}}{\sqrt{n}}}^{\tau,\epsilon}(f) \qquad \text{for all } f \in \mathcal{F}.$$

### 3.4 LIMITATIONS AND POTENTIAL EXTENSIONS

In the previous sections, we presented our results and their universality, to underline that they are widely applicable in machine learning. In this section, we discuss three relative limitations of our results: the assumption of the compactness of sample space, the assumption of finite Dudley entropy, and the expression of constants.

Compactness of the sample space $\Xi$ (Assumption 2.1.1) is essential to control worst-case distributions of the robust objective (2), given our level of generality. This assumption is in line with some recent studies (Azizian et al., 2023a; Blanchet & Shapiro, 2023); see also Gao et al. (2022) which uses bounded growth assumptions. Such assumptions are reasonable, as standard statistical frameworks involving Gaussian or heavy tail distributions could be covered by truncating.

In our study, considering loss families with finite Dudley's entropy (Assumption 2.1.3) is crucial to limit the dependence on the sample dimension. This assumption is satisfied for Lipschitz parametric losses with bounded parameter space, and it is not clear if a dimension-free generalization could be established for non-parametric losses. For instance, Zeng & Lam (2022) dealing with non-parametric losses, exhibits generalization guarantees with exponential dependence in the dimension.

Finally, regarding the generalization constants, we could improve them in several ways. For instance, leveraging the structure of specific models would allow to obtain estimates of the constants $\lambda_{\text{low}}$, $\rho_{\text{crit}}$; this is what we did for the linear models in Section 3.2. Taking into account the optimization procedure which selects a small set of solutions could also be interesting in order to have sharper constants on the class $\mathcal{F}$.

## 4 SKETCH OF THE PROOF

This section presents our strategy to prove the generalization results of Section 3 (Theorems 3.1 and 3.2). The strength of our approach is to use flexible nonsmooth analysis arguments, able to cover the general situation of arbitrary (continuous) cost and objective functions. We present the main approach in Section 4.1, based on a duality formula and a lower bound $\lambda_{\text{low}}$ on the dual variable. In Section 4.2, we focus on the latter and shed lights on its role. Finally, we explain in Section 4.3 the extension to regularized models.

### 4.1 MAIN APPROACH

Compared to the original formulation (4), the dual representation significantly diminishes the problem's degrees of freedom, and is usually the starting point of most studies of WDRO; see e.g. Kuhn et al. (2019); Azizian et al. (2023b). Given any distribution $Q \in \mathcal{P}(\Xi)$ and radius $\rho > 0$, it holds

$$R_{\rho,Q}(f) = \inf_{\lambda \geq 0} \{\lambda\rho + \mathbb{E}_{\xi \sim Q}[\phi(\lambda, f, \xi)]\},$$

where the *dual generator* $\phi$ is a convex function with respect to $\lambda$, and Lipschitz continuous with respect to $f$. For Wasserstein robust models, $\phi$ has the expression (see e.g. Blanchet & Murthy (2019))

$$\phi(\lambda, f, \xi) = \sup_{\zeta \in \Xi} \{f(\zeta) - \lambda c(\xi, \zeta)\}.$$

Observe that $\phi$ is naturally convex in $\lambda$, but also nonsmooth. The originality of our approach is to build on this nonsmoothness by using a rationale of nonsmooth analysis, which allows us to cover the case of other dual generators as for the regularized versions; see next section.

Let us then outline the main steps to establish Theorem 3.1:

1. **Dual lower bound.** Given $\beta > 0$ appearing in Theorem 3.1, We establish the existence of a dual lower bound $\lambda_{\text{low}} > 0$, which holds with high probability for all $f \in \mathcal{F}$, for a small enough radius $\rho \le \frac{\rho_{\text{crit}}}{4} - \frac{\beta}{\sqrt{n}}$:

$$\widehat{R}_\rho(f) = \inf_{\lambda \in [\lambda_{\text{low}}, \infty)} \left\{ \lambda\rho + \mathbb{E}_{\xi \sim \widehat{P}_n}[\phi(\lambda, f, \xi)] \right\}.$$

   This is done in Appendix E.1.

2. **Concentration of the radius.** Let us write for all $\lambda \ge \lambda_{\text{low}}$ and $f \in \mathcal{F}$

$$\lambda\rho + \mathbb{E}_{\widehat{P}_n}[\phi(\lambda, f)] \ge \lambda \left( \rho - \left( \frac{\mathbb{E}_P[\phi(\lambda, f)] - \mathbb{E}_{\widehat{P}_n}[\phi(\lambda, f)]}{\lambda} \right) \right) + \mathbb{E}_P[\phi(\lambda, f)]$$

$$\ge \lambda(\rho - \alpha_n) + \mathbb{E}_P[\phi(\lambda, f)], \tag{9}$$

   where we define the uniform gap $\alpha_n$ by

$$\alpha_n = \sup \left\{ \mathbb{E}_P[\mu\phi(\mu^{-1}, f)] - \mathbb{E}_{\widehat{P}_n}[\mu\phi(\mu^{-1}, f)] : (\mu, f) \in (0, \lambda_{\text{low}}^{-1}] \times \mathcal{F} \right\}. \tag{10}$$

   This quantity can be bounded with high probability by $\frac{\alpha}{\sqrt{n}}$ – where $\alpha > 0$ is the constant from Theorem 3.1. To obtain such a bound, we rely on known uniform concentration theorems for Lipschitz functions (Boucheron et al., 2013). Concentration constants are computed in Appendix C.

3. **Generalization bound.** We can now obtain the result. Taking the infimum over $\lambda \ge \lambda_{\text{low}}$ in (9), we obtain with high probability for all $f \in \mathcal{F}$,

$$\widehat{R}_\rho(f) \ge R_{\rho - \alpha/\sqrt{n}}(f) \ge \mathbb{E}_{\xi \sim P}[f(\xi)],$$

   whenever $\frac{\alpha}{\sqrt{n}} < \rho \le \frac{\rho_{\text{crit}}}{4} - \frac{\beta}{\sqrt{n}}$. This interval is nonempty if $n > 16(\alpha + \beta)^2/\rho_{\text{crit}}^2$. Since $\widehat{R}_\rho(f)$ is non-decreasing with respect to $\rho$, we have $\widehat{R}_\rho(f) \ge \mathbb{E}_{\xi \sim P}[f(\xi)]$ for any $\rho > \frac{\alpha}{\sqrt{n}}$ as long as $n > 16(\alpha + \beta)^2/\rho_{\text{crit}}^2$.

## 4.2 Definition of the lower bound

$\lambda_{\text{low}}$ defines a dual lower bound on the true risk $R_\rho(f)$, making it independent from samples randomness. In our proof, we then show that this lower bound holds with high probability on the empirical robust risk $\widehat{R}_\rho(f)$ using the convexity of $\phi$. This is done in Proposition E.2 in Appendix.

We now explain the definition of $\lambda_{\text{low}}$ more precisely. We consider the *maximal radius* function

$$\rho_{\text{max}}(\lambda) = \inf_{f \in \mathcal{F}} \mathbb{E}_{\xi \sim P}[-\partial_\lambda^+ \phi(\lambda, f, \xi)].$$

At a given $\lambda$, this function indicates the maximum value $\rho$ can take for the dual solution of $R_\rho(f)$ to be

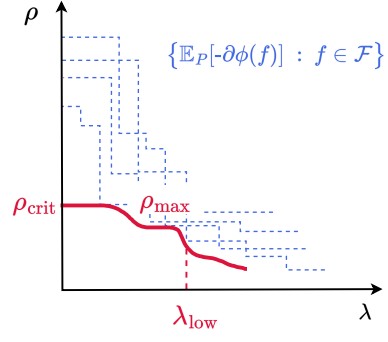

Figure 1: A central object of our analysis: the maximal radius $\rho_{\text{max}}$, defined from the lower envelope of derivatives of $\phi$.

higher than $\lambda$. In particular, by convexity of $\phi$, we can easily verify that: if $\rho \leq \rho_{\max}(\lambda)$ for all $\lambda \in [0, 2\lambda_{\text{low}}]$, then the dual bound $2\lambda_{\text{low}}$ holds on the true robust risk, and we have:

$$R_\rho(f) = \inf_{\lambda \geq 2\lambda_{\text{low}}} \{\lambda \rho + \mathbb{E}_{\xi \sim P}[\phi(\lambda, f, \xi)]\}. \tag{11}$$

As illustrated by Figure 1, $\rho_{\max}$ reaches its highest value at zero, which is actually the *critical radius*, $\rho_{\text{crit}}$. The crux of the proof is to show there exists a value $\lambda_{\text{low}}$ allowing to choose radius values of order $\rho_{\text{crit}}$. This comes by passing to the limit in $\rho_{\max}(\lambda)$.

**Lemma 4.1.** *We have* $\lim_{\lambda \to 0^+} \rho_{\max}(\lambda) = \rho_{\text{crit}}$. *In particular, there exists* $\lambda_{\text{low}} > 0$ *such that if* $\rho \leq \frac{\rho_{\text{crit}}}{4}$, *then* (11) *holds for all* $f \in \mathcal{F}$.

Such a result may be surprising since $\phi$ is a nonsmooth function and $\rho_{\max}$, defined from the lower envelope of (discontinuous) derivatives of $\phi$ is in general highly discontinuous. In order to establish it, we use tools from nonsmooth analysis (Appendix A.1) and leverage compactness of the class $\mathcal{F}$.

### 4.3 EXTENSION TO (DOUBLE) REGULARIZATION

The strategy of Section 4.1 is flexible enough to be extended to the regularized setting of Section 3.3. Indeed, the regularized problem also has a dual representation, with a dual generator defined by

$$\phi^{\tau,\epsilon}(\lambda, f, \xi) = (\epsilon + \lambda\tau) \log \mathbb{E}_{\zeta \sim \pi_0(\cdot|\xi)} \left[ e^{\frac{f(\zeta) - \lambda c(\xi, \zeta)}{\epsilon + \lambda\tau}} \right],$$

where $\epsilon > 0$ and $\tau \geq 0$. Strong duality has been shown in Azizian et al. (2023b). We explain in Appendix B, Proposition B.2 how it applies to our general setting. This regularized dual generator leads to a smooth counterpart of the key function $\rho_{\max}$ from the proof and to the regularized critical radius (8). In particular, we can show the regularized version of Lemma 4.1.

**Lemma 4.2.** $\lim_{\lambda \to 0^+} \rho_{\max}^{\tau,\epsilon}(\lambda) = \rho_{\text{crit}}^{\tau,\epsilon}$. *In particular, there exists* $\lambda_{\text{low}}^{\tau,\epsilon} > 0$ *such that if* $\rho \leq \frac{\rho_{\text{crit}}^{\tau,\epsilon}}{4}$,

$$R_\rho^{\tau,\epsilon}(f) = \inf_{\lambda \geq 2\lambda_{\text{low}}^{\tau,\epsilon}} \{\lambda \rho + \mathbb{E}_{\xi \sim P}[\phi^{\tau,\epsilon}(\lambda, f, \xi)]\} \qquad \text{for all } f \in \mathcal{F}.$$

Then we obtain Theorem 3.2 by repeating the proof scheme of Section 4.1. The core results that simultaneously lead to Theorems 3.1 and 3.2 are gathered in Appendix E.1. Due to the smoothness of $\rho_{\max}^{\tau,\epsilon}$ an expression of $\lambda_{\text{low}}^{\tau,\epsilon}$ can also be obtained; see Lemma D.2.

The key difference brought by regularization is that Lipschitzness of $\mu\phi(\mu^{-1}, f, \xi)$ is lost when $\mu \to 0$. This prevents us from using the concentration result – essential to bound the gap $\alpha_n$ (10) – unless we can set a lower bound on $\mu$, or equivalently an upper bound on $\lambda$. This issue is inherent to the regularized setting and may occur over the whole family $\mathcal{F}$ and the space $\Xi$; we provide an example in Proposition F.4 to illustrate this. The next lemma aims to overcome this issue by establishing the existence of such an upper bound for any distribution (see Lemma D.3 for a proof).

**Lemma 4.3.** *Let* $Q \in \mathcal{P}(\Xi)$, $\rho > m_c$ *and* $\lambda_{\text{up}} := \frac{2\|\mathcal{F}\|_\infty}{\rho - m_c}$. *Then for all* $f \in \mathcal{F}$,

$$R_{\rho,Q}^{\tau,\epsilon}(f) = \inf_{\lambda \in [0, \lambda_{\text{up}}]} \{\lambda \rho + \mathbb{E}_{\xi \sim Q}[\phi^{\tau,\epsilon}(\lambda, f, \xi)]\}.$$

## 5 CONCLUDING REMARKS

In this work, we provide exact generalization guarantees of (regularized) Wasserstein robust models, without restrictive assumptions (on the transport cost or the class of functions). We achieve these universal results by directly addressing the intrinsic nonsmoothness of robust problems. Our results thus give users freedom when choosing the radius $\rho$: it is not necessary to consider specific regimes for $\rho$ in order to expect good generalization from robust models. Further research can now focus on practical aspects: it would be of premier interest to design efficient procedures for selecting $\rho$, and more generally, scalable algorithms for solving distributionally robust optimization problems.

### ACKNOWLEDGMENTS

This research was partially supported by MIAI@Grenoble Alpes (ANR-19-P3IA-0003).

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

# Supplemental material

This supplemental gathers recalls, technical lemmas and definitions, examples, and detailed proofs of the results from the main text. The core of our contributions is presented in Appendices D and E. The whole supplemental is organized as follows:

– In Appendix A, we recall some essential mathematical tools. They include continuity notions in nonsmooth analysis, the envelope formula to differentiate supremum functions (Theorem A.1) and a uniform concentration inequality (Theorem A.2).

– In Appendix B, we present strong duality results for WDRO and its regularized version. We explain in particular how the duality theorem from Azizian et al. (2023b) can be easily adapted to our setting.

– Appendix C contains preliminary computations in view of applying the uniform concentration theorem.

– In Appendix D, we demonstrate the existence of a dual lower bound in the standard and regularized cases. In particular, the proofs involve the maximal radius introduced in Section 4.2.

– By using these preliminary results, in Appendix E, we prove our main results. They include the generalization theorems (Theorem 3.1 and 3.2), the excess bounds (Proposition 3.1 and Proposition 3.3) and the constants of the linear models (Proposition 3.2).

– Appendix F contains results supporting several remarks of the main. They include the interpretation of the critical radius in the regularized case, a counter-example justifying the upper bound in the regularized case and the interpretation of the restrictive compactness assumptions used in Azizian et al. (2023a).

NOTATIONS

Throughout the proofs will use the following notations:

**On probability spaces.** Given the metric space $(\Xi, d)$ where $d$ is a distance, we denote the space of probability measures on $\Xi$ by $\mathcal{P}(\Xi)$. For all $\pi \in \mathcal{P}(\Xi \times \Xi)$, $i \in \{1, 2\}$, we denote the $i^{\text{th}}$ marginal of $\pi$ by $[\pi]_i$. We denote the Dirac mass at $\xi \in \Xi$ by $\delta_\xi$. Given a measurable function $g : \Xi \to \mathbb{R}$, we denote the expectation of $g$ with respect to $Q \in \mathcal{P}(\Xi)$ by $\mathbb{E}_{\xi \sim Q}[g(\xi)]$ and we may also use the shorthand $\mathbb{E}_Q[g]$. For $\pi \in \mathcal{P}(\Xi)$, we use the notation $\pi^g$ for the Gibbs distribution $\mathrm{d}\pi^g \propto e^g \mathrm{d}\pi$.

**On function spaces.** For a continuous function $f : \Xi \to \mathbb{R}$, we denote the uniform norm by $\|f\|_\infty = \sup_{\xi \in \Xi} |f(\xi)|$. By extension, if $\mathcal{F}$ is a set of functions we denote $\|\mathcal{F}\|_\infty := \sup_{f \in \mathcal{F}} \|f\|_\infty$. Whenever well-defined, we denote the set of maximizers of $f$ on $\Xi$ by $\arg\max_\Xi f := \{\zeta \in \Xi : f(\zeta) = \max_{\xi \in \Xi} f(\xi)\}$. Given a metric space $(\mathcal{X}, \mathrm{dist})$, we say a function $h : \mathcal{X} \to \mathbb{R}$ is *Lipschitz* with constant $L > 0$ if for all $x, y \in \mathcal{X}$, $|h(x) - h(y)| \leq L \, \mathrm{dist}(x, y)$. For $\phi \colon \mathbb{R} \times \mathcal{X} \to \mathbb{R}$, we denote $\partial_\lambda^+ \phi$ the right-sided derivative with respect to $\lambda \in \mathbb{R}$, and $\partial_\lambda \phi$ the derivative, whenever well-defined.

**In Wasserstein robust models.**

- $\phi(\lambda, f, \xi) = \sup_{\zeta \in \Xi} \{f(\zeta) - \lambda c(\xi, \zeta)\}$
- $\psi(\mu, f, \xi) = \mu \phi(\mu^{-1}, f, \xi)$
- $\rho_{\mathrm{crit}} = \inf_{f \in \mathcal{F}} \mathbb{E}_{\xi \sim P} [\min \{c(\xi, \zeta) : \zeta \in \arg\max_\Xi f\}].$

**In Wasserstein robust models with double regularization.**

- $\phi^{\tau, \epsilon}(\lambda, f, \xi) = (\epsilon + \lambda\tau) \log \mathbb{E}_{\zeta \sim \pi_0(\cdot|\xi)} \left[ e^{\frac{f(\zeta) - \lambda c(\xi, \zeta)}{\epsilon + \lambda\tau}} \right]$
- $\psi^{\tau, \epsilon}(\mu, f, \xi) = \mu \phi^{\tau, \epsilon}(\mu^{-1}, f, \xi)$

- $\rho_{\mathrm{crit}}^{\tau,\epsilon} = \inf_{f \in \mathcal{F}} \mathbb{E}_{\xi \sim P} \left[ \mathbb{E}_{\zeta \sim \pi_0^{f/\epsilon}(\cdot|\xi)} \left[ \frac{\tau}{\epsilon} f(\zeta) + c(\xi, \zeta) \right] - \tau \log \mathbb{E}_{\zeta \sim \pi_0(\cdot|\xi)} \left[ e^{\frac{f(\zeta)}{\epsilon}} \right] \right].$

# A    RECALLS AND TECHNICAL PRELIMINARIES

## A.1    NONSMOOTH ANALYSIS

In this part, we use the notation $G : \mathcal{X} \rightrightarrows \mathcal{Y}$ to denote a function $G$ defined on $\mathcal{X}$ and valued in the set of subsets of $\mathcal{Y}$.

Semicontinuity notions will be necessary to understand the proof of Lemma D.1. They are regularity notions recurrently arising when manipulating nonsmooth convex functions.

**Definition A.1** (Lower and upper semicontinuity (Aliprantis & Border, 2006, 2.42)). *Let $(\mathcal{X}, \mathrm{dist})$ be a metric space and let $f : \mathcal{X} \rightarrow \mathbb{R}$. Then*

1. *$f$ is called* lower semicontinuous *if for all $x \in \mathcal{X}$, $\liminf_{y \to x} f(y) \geq f(x)$.*

2. *$f$ is called* upper semicontinuous *if for all $x \in \mathcal{X}$, $\limsup_{y \to x} f(y) \leq f(x)$.*

*In particular, if $f$ is lower semicontinuous, then $-f$ is upper semicontinuous.*

*Outer semicontinuity* can be seen as the set-valued counterpart of upper semicontinuity:

**Definition A.2** (Outer semicontinuity). *Let $\mathcal{X}$ and $\mathcal{Y}$ two metric spaces. Then a measurable and compact-valued map $G : \mathcal{X} \rightrightarrows \mathcal{Y}$ is called* outer semicontinuous *at $x \in \mathcal{X}$ if for all open subset $V \subset \mathcal{Y}$ containing $G(x)$, there exists a neighborhood $U$ of $x$ which is such that for all $w \in U$, $G(w) \subset V$.*

Semicontinuity of maximum and $\arg\max$ functions are central to the proof of Lemma D.1:

**Lemma A.1** (Semicontinuity of maximum value (Aliprantis & Border, 2006, 17.30)). *Let $\mathcal{X}$ and $\Xi$ be two metric spaces and let $G : \mathcal{X} \rightrightarrows \Xi$ be outer semicontinuous with nonempty compact values, $h : \Xi \times \Xi \rightarrow \mathbb{R}$ continuous. Then the function*

$$x \mapsto \max\{h(x, v) \ : \ v \in G(x)\}$$

*is upper semicontinuous. In particular, $u \mapsto \min\{h(u, v) \ : \ v \in G(x)\}$ is lower semicontinuous.*

**Lemma A.2** (Semicontinuity of maximizers (Aliprantis & Border, 2006, 17.31)). *If $\mathcal{X}$ is a metric space, $(\Xi, d)$ is a compact metric space, and $h : \mathcal{X} \times \Xi \rightarrow \mathbb{R}$ is continuous, then the function $x \mapsto \max_{z \in \Xi} h(x, z)$ is continuous, and the set-valued map $x \mapsto \arg\max_{z \in \Xi} h(x, z)$ is outer semicontinuous.*

We recall the definition of gradient for a nonsmooth convex function. This the *subdifferential*.

**Definition A.3** (Subdifferential of convex function). *Let $\phi : \mathbb{R}^m \rightarrow \mathbb{R}$ be a convex function. Then we call subdifferential of $\phi$ the set-valued map $\partial\phi : \mathbb{R}^m \rightrightarrows \mathbb{R}^m$ such that for all $x \in \mathbb{R}^m$ and $y \in \mathbb{R}^m$,*

$$\phi(y) \geq \phi(x) + \langle v, y - x \rangle \text{ for all } v \in \partial\phi(x).$$

In particular, we may apply the envelope formula to compute the subdifferential of a maximum function:

**Theorem A.1** (Envelope formula (Clarke, 1990, Cor. 1, Chapter 2.8)). *Let $(\Xi, d)$ be a compact metric space and $g : \mathbb{R}^m \times \Xi \rightarrow \mathbb{R}$ such that*

1. *For all $x \in \mathbb{R}^m$, $g(x, \cdot)$ is continuous.*

2. *For all $\zeta \in \Xi$, $g(\cdot, \zeta)$ is convex with subdifferential $\partial_x g(\cdot, \zeta)$.*

*Then $G := \sup_{\zeta \in \Xi} g(\cdot, \zeta)$ is convex on $\mathbb{R}^m$, and its subdifferential is given for all $x \in \mathbb{R}^m$ by*

$$\partial G(x) := \mathrm{conv}\{v \ : \ v \in \partial_x g(x, \zeta), \ \zeta \in \arg\max_{\Xi} g(x, \cdot)\}.$$

*where $\mathrm{conv}$ denotes the convex hull of a set.*

### A.2 Uniform concentration inequality

We recall concentration inequalities that gives a high probability uniform bound for a family of bounded and Lipschitz functions. We refer the reader to Boucheron et al. (2013) for a complete reference on concentration inequalities, and Lemma G.2 in Azizian et al. (2023a) for the proof of such a result.

**Theorem A.2** (Uniform concentration (Azizian et al., 2023a, Lem. G.2)). *Let $(\mathcal{X}, \mathrm{dist})$ be a (totally bounded) separable metric space, $P$ a probability distribution on a probability space $\Xi$, and $\widehat{P}_n = \frac{1}{n}\sum_{i=1}^{n}\delta_{\xi_i}$ with $\xi_1, \ldots, \xi_n \overset{i.i.d.}{\sim} P$. Consider a measurable mapping $X : \mathcal{X} \times \Xi \to \mathbb{R}$ and assume that,*

*(i) There is a constant $L > 0$ such that, for each $\xi \in \Xi$, $x \mapsto X(x,\xi)$ is $L$-Lipschitz.*

*(ii) $X(\cdot, \xi)$ almost surely belongs to $[a, b]$.*

*Then, for any $\delta \in (0,1)$, we respectively have*

*1. With probability at least $1 - \delta$,*

$$\sup_{x \in \mathcal{X}} \left\{ \mathbb{E}_{\xi \sim \widehat{P}_n}[X(x,\xi)] - \mathbb{E}_{\xi \sim P}[X(x,\xi)] \right\} \leq \frac{48 L \mathcal{I}(\mathcal{X}, \mathrm{dist})}{\sqrt{n}} + (b-a)\sqrt{2\frac{\log \frac{1}{\delta}}{n}}.$$

*2. With probability at least $1 - \delta$,*

$$\sup_{x \in \mathcal{X}} \left\{ \mathbb{E}_{\xi \sim P}[X(x,\xi)] - \mathbb{E}_{\xi \sim \widehat{P}_n}[X(x,\xi)] \right\} \leq \frac{48 L \mathcal{I}(\mathcal{X}, \mathrm{dist})}{\sqrt{n}} + (b-a)\sqrt{2\frac{\log \frac{1}{\delta}}{n}}.$$

The quantity $\mathcal{I}(\mathcal{X}, \mathrm{dist})$ is defined as follows:

**Definition A.4.** *Given a compact metric space $(\mathcal{X}, \mathrm{dist})$, Dudley's entropy integral, $\mathcal{I}(\mathcal{X}, \mathrm{dist})$, is defined as*

$$\mathcal{I}(\mathcal{X}, \mathrm{dist}) := \int_0^\infty \sqrt{\log N(t, \mathcal{X}, \mathrm{dist})} dt$$

*where $N(t, \mathcal{X}, \mathrm{dist})$ denotes the $t$-packing number of $\mathcal{X}$, which is the maximal number of points in $\mathcal{X}$ that are at least at a distance $t$ from each other.*

We may recall some properties of Dudley's entropy for Cartesian products and segments from $\mathbb{R}$. These are known results, see e.g. Wainwright (2019) and Lemmas G.3 and G.4 from Azizian et al. (2023a) for proofs.

**Lemma A.3** (Dudley's integral estimates).

*1. (on Cartesian products) Let $(\mathcal{X}_1, \mathrm{dist}_1)$ and $(\mathcal{X}_2, \mathrm{dist}_2)$ be two metric spaces. Consider the product space $\mathcal{X} := \mathcal{X}_1 \times \mathcal{X}_2$ equipped with the distance $\mathrm{dist} := \mathrm{dist}_1 + \mathrm{dist}_2$. Then we have the inequality*

$$\mathcal{I}(\mathcal{X}, \mathrm{dist}) \leq \mathcal{I}(\mathcal{X}_1, \mathrm{dist}_1) + \mathcal{I}(\mathcal{X}_2, \mathrm{dist}_2).$$

*2. (on $\mathbb{R}$) Let $c > 0$. Then we have the inequality*

$$\mathcal{I}([0, c], |\cdot|) \leq \frac{3c}{2}.$$

## B Strong duality

In this section, we recall duality results for WDRO Blanchet & Murthy (2019); Gao & Kleywegt (2023); Zhang et al. (2022) and its regularized version Azizian et al. (2023b). We recall the optimal transport cost with cost $c$ for $(Q, Q') \in \mathcal{P}(\Xi) \times \mathcal{P}(\Xi)$:

$$W_c(Q, Q') = \inf \left\{ \mathbb{E}_{(\xi,\zeta) \sim \pi}[c(\xi, \zeta)] \ : \ \pi \in \mathcal{P}(\Xi \times \Xi), [\pi]_1 = Q, [\pi]_2 = Q' \right\}.$$

**Proposition B.1** (Strong duality, standard WDRO). *Under Assumption 2.1, for any $Q \in \mathcal{P}(\Xi)$ and $\rho > 0$, then*

$$\sup_{W_c(Q,Q') \leq \rho} \mathbb{E}_{\xi \sim Q'}[f(\xi)] = \inf_{\lambda \geq 0} \left\{ \lambda \rho + \mathbb{E}_{\xi \sim Q}[\phi(\lambda, f, \xi)] \right\}.$$

*Proof.* This is an application of Theorem 1 from Blanchet & Murthy (2019). In particular, Assumptions 1 and 2 from Blanchet & Murthy (2019) are satisfied through Assumption 2.1. □

**Proposition B.2** (Strong duality, regularized WDRO). *Under Assumption 2.1, for any $Q \in \mathcal{P}(\Xi)$ and $\rho > 0$, if there exists $\pi \in \mathcal{P}(\Xi \times \Xi)$ such that $\mathbb{E}_\pi[c] + \tau D_{\mathrm{KL}}(\pi \| \pi_0) < \rho$, then*

$$\sup_{\substack{\pi \in \mathcal{P}(\Xi \times \Xi), [\pi]_1 = Q \\ \mathbb{E}_\pi[c] + \tau \mathrm{KL}(\pi \| \pi_0) \leq \rho}} \left\{ \mathbb{E}_{\xi \sim [\pi]_2}[f(\zeta)] - \epsilon \mathrm{KL}(\pi \| \pi_0) \right\} = \inf_{\lambda \geq 0} \left\{ \lambda \rho + \mathbb{E}_{\xi \sim Q}[\phi^{\tau,\epsilon}(\lambda, f, \xi)] \right\}. \quad (12)$$

*In particular, if $\rho > m_c$, (12) holds.*

*Proof.* This is an application of Theorem 3.1 from Azizian et al. (2023b), which is a corollary to Theorem 2.1 Azizian et al. (2023b). In particular, if $\rho > m_c$ the coupling $\pi_0$ satisfies $\mathbb{E}_{\pi_0}[c] + \tau D_{\mathrm{KL}}(\pi_0 \| \pi_0) = \mathbb{E}_{\pi_0}[c] \leq m_c < \rho$.

Note that the proofs of Theorems 2.1 and 3.1 from Azizian et al. (2023b) can be easily extended to a general compact metric space $(\Xi, d)$, without being rewritten entirely. Precisely, only two arguments in their proofs rely on the real-valued setting Rockafellar & Wets (1998) but can be directly extended to a general metric space as follows:

- In the proof of Theorem 2.1 from Azizian et al. (2023b), one needs to justify

$$\sup \left\{ \mathbb{E}_{\xi \sim P} \left[ \varphi(\xi, \zeta(\xi)) \right] \; : \; \zeta : \Xi \to \Xi \text{ measurable} \right\} \geq \mathbb{E}_{\xi \sim P} \left[ \sup_{\zeta \in \Xi} \varphi(\xi, \zeta) \right]. \quad (13)$$

  To this end, the authors use the notion of normal integrand from Rockafellar & Wets (1998). Actually, (13) holds true in a compact metric space: if $\varphi$ is continuous, then by compactness of $\Xi$, the set-valued map $\xi \mapsto \arg\max_{\zeta \in \Xi} \varphi(\xi, \zeta)$ admits a measurable selection $\zeta^*$, by the measurable maximum theorem, see 18.19 in Aliprantis & Border (2006). Such a selection $\zeta^*$ then satisfies $\varphi(\xi, \zeta^*(\xi)) = \sup_{\zeta \in \Xi} \varphi(\xi, \zeta)$ for all $\xi \in \Xi$, hence the result.

- In the proof of Theorem 3.1 from Azizian et al. (2023b), $g^\varphi = \sup_{\zeta \in \Xi} \varphi(\cdot, \zeta)$ is actually continuous by Lemma A.2 and the approximation by the infimal convolutions $(g_k^\varphi)_{k \in \mathbb{N}}$ need not be done.

Note that the convexity of $\Xi$ is not required (although stated in Assumption 1 from Azizian et al. (2023b)). □

## C  CONCENTRATION CONSTANTS

In this part, we compute some constants in view of applying Theorem A.2 for the main proofs of Appendix E.

### C.1  STANDARD WDRO

For standard WDRO, we compute bounds (i) and global Lipschitz constants (ii) for $\phi$ and $\psi$.

**Lemma C.1** (Concentration conditions for WDRO). *we have the following:*

1. *(i) For all $\lambda \geq 0$, $f \in \mathcal{F}$ and $\xi \in \Xi$, $\phi(\lambda, f, \xi) \in [-\|\mathcal{F}\|_\infty, \|\mathcal{F}\|_\infty]$.*
   *(ii) For all $\lambda \geq 0$ and $\xi \in \Xi$, $f \mapsto \phi(\lambda, f, \xi)$ is Lipschitz continuous on $\mathcal{F}$ with constant 1.*

2. *(i) Given $\lambda_{\mathrm{low}} > 0$, for all $\mu \in (0, \lambda_{\mathrm{low}}^{-1}]$ and $f \in \mathcal{F}$, $\psi(\mu, f, \xi) \in \left[ -\frac{\|\mathcal{F}\|_\infty}{\lambda_{\mathrm{low}}}, \frac{\|\mathcal{F}\|_\infty}{\lambda_{\mathrm{low}}} \right]$.*

*(ii) For all $\xi \in \Xi$, $(\mu, f) \mapsto \psi(\mu, f, \xi)$ is Lipschitz continuous on $(0, \lambda_{\text{low}}^{-1}]$ with constant $\|\mathcal{F}\|_\infty + \lambda_{\text{low}}^{-1}$.*

*Proof.* 1. (i) Let $(\lambda, f, \xi) \in \mathbb{R}_+ \times \mathcal{F} \times \Xi$. Recall that $\phi(\lambda, f, \xi) := \sup_{\zeta \in \Xi} \{f(\zeta) - \lambda c(\xi, \zeta)\}$. Since $c$ is nonnegative, we have $\phi(\lambda, f, \xi) \le \|\mathcal{F}\|_\infty$. On the other hand, since $c(\xi, \xi) = 0$, we also have $\phi(\lambda, f, \xi) \ge f(\xi) \ge -\|\mathcal{F}\|_\infty$. Finally, we have $\phi(\lambda, f, \xi) \in [-\|\mathcal{F}\|_\infty, \|\mathcal{F}\|_\infty]$.

(ii) Let $\lambda \ge 0$, $\xi \in \Xi$ and $(f, f') \in \mathcal{F} \times \mathcal{F}$. For all $\zeta \in \Xi$, we have

$$f(\zeta) - \lambda c(\xi, \zeta) - \phi(\lambda, f', \xi) \le f(\zeta) - \lambda c(\xi, \zeta) - (f'(\zeta) - \lambda c(\xi, \zeta))$$
$$\le f(\zeta) - f'(\zeta)$$
$$\le \|f - f'\|_\infty.$$

Taking the supremum over $\zeta \in \Xi$ on the left-hand side gives $\phi(\lambda, f, \xi) - \phi(\lambda, f', \xi) \le \|f - f'\|_\infty$. Permuting the roles of $f$ and $f'$ yields $|\phi(\lambda, f, \xi) - \phi(\lambda, f', \xi)| \le \|f - f'\|_\infty$. We proved that $\phi(\lambda, \cdot, \xi)$ is 1-Lipschitz continuous.

2. (i) Now, let $\lambda_{\text{low}} > 0$ and let $(\mu, f, \xi) \in (0, \lambda_{\text{low}}^{-1}] \times \mathcal{F} \times \Xi$ be arbitrary. Then we have

$$\mu \phi(\mu^{-1}, f, \xi) = \sup_{\zeta \in \Xi} \{\mu f(\zeta) - c(\xi, \zeta)\} \le \frac{\|\mathcal{F}\|_\infty}{\lambda_{\text{low}}}.$$

On the other hand, using $c(\xi, \xi) = 0$ we obtain

$$\sup_{\zeta \in \Xi} \{\mu f(\zeta) - c(\xi, \zeta)\} \ge \mu f(\xi) \ge -\frac{\|\mathcal{F}\|_\infty}{\lambda_{\text{low}}},$$

whence we have $\mu \phi(\mu^{-1}, f, \xi) \in \left[-\frac{\|\mathcal{F}\|_\infty}{\lambda_{\text{low}}}, \frac{\|\mathcal{F}\|_\infty}{\lambda_{\text{low}}}\right]$.

(ii) Toward a proof of 2. (ii), let $\lambda_{\text{low}} > 0$, and $\xi \in \Xi$ and $\mu \in (0, \lambda_{\text{low}}]$. Remark that $\mu \phi(\mu^{-1}, f, \xi) = \sup_{\zeta \in \Xi} \{\mu f(\zeta) - c(\xi, \zeta)\}$. The function $(\mu, f) \mapsto \mu \phi(\mu^{-1}, f, \xi)$ write as a composition $u \circ v$ where $u(h) := \sup_{\zeta \in \Xi} \{h(\zeta) - c(\xi, \zeta)\}$ for $h \in C(\Xi, \mathbb{R})$, and $v(\mu, f) := \mu f$ for $\mu \in (0, \lambda_{\text{low}}^{-1}]$. $u$ is 1-Lipschitz continuous with respect to $\|\cdot\|_\infty$. As to $v$, we can write

$$\mu f - \mu' f' = \mu(f - f') + f'(\mu - \mu'),$$

whence $v$ is clearly $(\|\mathcal{F}\|_\infty + \lambda_{\text{low}}^{-1})$-Lipschitz continuous on $(0, \lambda_{\text{low}}^{-1}] \times \mathcal{F}$. By composition, $u \circ v$ is Lipschitz continuous with constant $\|\mathcal{F}\|_\infty + \lambda_{\text{low}}^{-1}$. $\qquad \square$

## C.2 Regularized WDRO

We now compute the analogous constants of the regularized setting.

We will use the following convexity lemma repeatedly:

**Lemma C.2** ((Azizian et al., 2023a, Lem. G.7)). *Let $g : \Xi \to \mathbb{R}$ be a measurable bounded function and $Q \in \mathcal{P}(\Xi)$. Then one has the inequality*

$$\log \mathbb{E}_{\zeta \sim Q}\left[e^{g(\zeta)}\right] \le \frac{\mathbb{E}_{\zeta \sim Q}[g(\zeta) e^{g(\zeta)}]}{\mathbb{E}_{\zeta \sim Q}[e^{g(\zeta)}]}.$$

The following is the regularized version of Lemma C.1:

**Lemma C.3** (Concentration conditions for regularized WDRO). *Let $\xi \in \Xi$. Then*

1. *(i) For all $\lambda \ge 0$ and $f \in \mathcal{F}$, $\phi^{\tau,\epsilon}(\lambda, f, \xi) \in [-\|\mathcal{F}\|_\infty - \lambda m_c, \|\mathcal{F}\|_\infty]$.*
   *(ii) For all $\lambda \ge 0$, $f \mapsto \phi^{\tau,\epsilon}(\lambda, f, \xi)$ is Lipschitz continuous with constant 1.*

2. *(i) Given $\lambda_{\text{low}} > 0$, for all $\mu \in [\lambda_{\text{up}}^{-1}, \lambda_{\text{low}}^{-1}]$ and $f \in \mathcal{F}$, $\psi^{\tau,\epsilon}(\mu, f, \xi) \in \left[-\frac{\|\mathcal{F}\|_\infty}{\lambda_{\text{low}}} - m_c, \frac{\|\mathcal{F}\|_\infty}{\lambda_{\text{low}}}\right]$.*

*(ii) Given $\lambda_{\mathrm{up}} > 0$, $(\mu, f) \mapsto \psi^{\tau,\epsilon}(\mu, f, \xi)$ is Lipschitz continuous on $[\lambda_{\mathrm{up}}^{-1}, \lambda_{\mathrm{low}}^{-1}] \times \mathcal{F}$ with constant $\|\mathcal{F}\|_\infty + \lambda_{\mathrm{low}}^{-1} + \left(\frac{\lambda_{\mathrm{up}}\epsilon}{\epsilon + \lambda_{\mathrm{up}}\tau}\right) m_c$.*

*Proof.* 1. (i) Let $(\lambda, f, \xi) \in \mathbb{R}_+ \times \mathcal{F} \times \Xi$. For all $\zeta \in \Xi$, $e^{\frac{f(\zeta) - \lambda c(\xi, \zeta)}{\epsilon + \lambda\tau}} \leq e^{\frac{\|\mathcal{F}\|_\infty}{\epsilon + \lambda\tau}}$. This gives

$$\phi^{\tau,\epsilon}(\lambda, f, \xi) \leq (\epsilon + \lambda\tau) \log \mathbb{E}_{\zeta \sim \pi_0(\cdot|\xi)}\left[e^{\frac{\|\mathcal{F}\|_\infty}{\epsilon + \lambda\tau}}\right] = \|\mathcal{F}\|_\infty. \tag{14}$$

On the other hand, $e^{\frac{f(\zeta) - \lambda c(\xi, \zeta)}{\epsilon + \lambda\tau}} \geq e^{\frac{-\|\mathcal{F}\|_\infty - \lambda c(\xi, \zeta)}{\epsilon + \lambda\tau}}$, which gives

$$\phi^{\tau,\epsilon}(\lambda, f, \xi) \geq (\epsilon + \lambda\tau) \log\left(e^{-\frac{\|\mathcal{F}\|_\infty}{\epsilon + \lambda\tau}} \mathbb{E}_{\zeta \sim \pi_0(\cdot|\xi)}\left[e^{-\frac{\lambda c(\xi, \zeta)}{\epsilon + \lambda\tau}}\right]\right)$$

$$\geq -\|\mathcal{F}\|_\infty + (\epsilon + \lambda\tau) \log \mathbb{E}_{\zeta \sim \pi_0(\cdot|\xi)}\left[e^{-\frac{\lambda c(\xi, \zeta)}{\epsilon + \lambda\tau}}\right]$$

$$\geq -\|\mathcal{F}\|_\infty - \lambda m_c, \tag{15}$$

where for the last inequality we used Jensen's inequality on the convex function $s \mapsto e^{-\frac{\lambda s}{\epsilon + \lambda\tau}}$.

Combining (14) and (15) gives

$$\phi^{\tau,\epsilon}(\lambda, f, \xi) \in [-\|\mathcal{F}\|_\infty - \lambda m_c, \|\mathcal{F}\|_\infty].$$

(ii) Let $\xi \in \Xi$ and $\lambda \geq 0$. To compute the Lipschitz constant of $f \mapsto \phi^{\tau,\epsilon}(\lambda, f, \xi)$, we compute the derivative of $h_v : t \mapsto \phi^{\tau,\epsilon}(\lambda, f + tv, \xi)$ where $t \in \mathbb{R}$ and for an arbitrary direction $v \in \mathcal{F}$. We have

$$h_v(t) = (\epsilon + \lambda\tau) \log \mathbb{E}_{\zeta \sim \pi_0(\cdot|\xi)}\left[e^{\frac{f(\zeta) + tv(\zeta) - \lambda c(\xi, \zeta)}{\epsilon + \lambda\tau}}\right].$$

It is easy to verify that $h_v'(t) = \mathbb{E}_{\zeta \sim \pi_0 \frac{f + tv - \lambda c(\xi, \cdot)}{\epsilon + \lambda\tau}(\cdot|\xi)}[v(\zeta)]$, whence $|h_v'(t)| \leq \|v\|_\infty$. This means that $\phi^{\tau,\epsilon}(\lambda, \cdot, \xi)$ has Lipschitz constant 1.

2. (i) Let $\lambda_{\mathrm{low}} > 0$ and $(\mu, f, \xi) \in (0, \lambda_{\mathrm{low}}^{-1}] \times \mathcal{F} \times \Xi$. $\lambda$. We deduce from (14) and (15), with $\lambda = \mu^{-1}$, that

$$\mu \phi^{\tau,\epsilon}(\mu^{-1}, f, \xi) \in \left[-\frac{\|\mathcal{F}\|_\infty}{\lambda_{\mathrm{low}}} - m_c, \frac{\|\mathcal{F}\|_\infty}{\lambda_{\mathrm{low}}}\right].$$

(ii) Now, let $\xi \in \Xi$. Our goal is to compute a Lipschitz constant of $(\mu, f) \mapsto \mu \phi^{\tau,\epsilon}(\mu^{-1}, f, \xi)$ on $[\lambda_{\mathrm{up}}^{-1}, \lambda_{\mathrm{low}}^{-1}] \times \mathcal{F}$. We first compute a Lipschitz constant of

$$h_f : \mu \mapsto \mu \phi^{\tau,\epsilon}(\mu^{-1}, f, \xi) = (\mu\epsilon + \tau) \log \mathbb{E}_{\zeta \sim \pi_0(\cdot|\xi)}\left[e^{\frac{\mu f(\zeta) - c(\xi, \zeta)}{\mu\epsilon + \tau}}\right]$$

on $[\lambda_{\mathrm{up}}^{-1}, \lambda_{\mathrm{low}}^{-1}]$, for an arbitrary $f \in \mathcal{F}$. The derivative of $h_f$ is

$$h_f'(\mu) = \frac{1}{\mu\epsilon + \tau} \frac{\mathbb{E}_{\zeta \sim \pi_0(\cdot|\xi)}\left[(\epsilon c(\xi, \zeta) + \tau f(\zeta)) e^{\frac{\mu f(\zeta) - c(\xi, \zeta)}{\mu\epsilon + \tau}}\right]}{\mathbb{E}_{\zeta \sim \pi_0(\cdot|\xi)}\left[e^{\frac{\mu f(\zeta) - c(\xi, \zeta)}{\mu\epsilon + \tau}}\right]} + \epsilon \log \mathbb{E}_{\zeta \sim \pi_0(\cdot|\xi)}\left[e^{\frac{\mu f(\zeta) - c(\xi, \zeta)}{\mu\epsilon + \tau}}\right],$$

which we write

$$h_f'(\mu) = \mathbb{E}_{\pi_0 \frac{\mu f - c(\xi, \cdot)}{\mu\epsilon + \tau}(\cdot|\xi)}\left[\frac{\epsilon c(\xi, \zeta) + \tau f(\zeta)}{\mu\epsilon + \tau}\right] + \epsilon \log \mathbb{E}_{\zeta \sim \pi_0(\cdot|\xi)}\left[e^{\frac{\mu f(\zeta) - c(\xi, \zeta)}{\mu\epsilon + \tau}}\right]. \tag{16}$$

We bound $h_f'(\mu)$ above. By Lemma C.2 with $Q = \pi_0(\cdot|\xi)$ and $g = \frac{\mu f - c(\xi, \cdot)}{\mu\epsilon + \tau}$, we have that

$$\epsilon \log \mathbb{E}_{\zeta \sim \pi_0(\cdot|\xi)}\left[e^{\frac{\mu f(\zeta) - c(\xi, \zeta)}{\mu\epsilon + \tau}}\right] \leq \mathbb{E}_{\zeta \sim \pi_0 \frac{\mu f - c(\xi, \cdot)}{\mu\epsilon + \tau}(\cdot|\xi)}\left[\frac{\epsilon\mu f(\zeta) - \epsilon c(\xi, \zeta)}{\mu\epsilon + \tau}\right]$$

which gives $h_f'(\mu) \leq \mathbb{E}_{\zeta \sim \pi_0 \frac{\mu f - c(\xi, \cdot)}{\mu\epsilon + \tau}}[f(\zeta)] \leq \|\mathcal{F}\|_\infty$.

Now we bound $h'_f(\mu)$ below. We start with the first term in (16). Since $c$ is nonnegative, we clearly have

$$\mathbb{E}_{\pi_0^{\frac{\mu f - c(\xi, \cdot)}{\mu \epsilon + \tau}}(\cdot|\xi)} \left[ \frac{\epsilon c(\xi, \zeta) + \tau f(\zeta)}{\mu \epsilon + \tau} \right] \geq \frac{-\tau \|\mathcal{F}\|_\infty}{\mu \epsilon + \tau} \tag{17}$$

As to the second term of (16), we have by Jensen's inequality,

$$\epsilon \log \mathbb{E}_{\zeta \sim \pi_0(\cdot|\xi)} \left[ e^{\frac{\mu f(\zeta) - c(\xi, \zeta)}{\mu \epsilon + \tau}} \right] \geq -\frac{\epsilon \mu \|\mathcal{F}\|_\infty}{\mu \epsilon + \tau} - \frac{\epsilon m_c}{\mu \epsilon + \tau} \tag{18}$$

Combining (17) and (18) gives $h'_f(\mu) \geq -\|\mathcal{F}\|_\infty - \frac{\lambda_{\mathrm{up}} \epsilon m_c}{\epsilon + \lambda_{\mathrm{up}} \tau}$. Finally, $h_f$ has Lipschitz constant $\|\mathcal{F}\|_\infty + \frac{\lambda_{\mathrm{up}} m_c}{\epsilon + \lambda_{\mathrm{up}} \tau}$

Since $\phi^{\tau, \epsilon}(\mu^{-1}, \cdot, \xi)$ has Lipschitz constant 1, then $\mu \leq \lambda_{\mathrm{low}}^{-1}$, the function $\mu \phi^{\tau, \epsilon}(\mu^{-1}, \cdot, \xi)$ has Lipschitz constant $\lambda_{\mathrm{low}}^{-1}$.

Now, we can obtain a Lipschitz constant for

$$h : (\mu, f) \mapsto (\mu \epsilon + \tau) \log \mathbb{E}_{\zeta \sim \pi_0(\cdot|\xi)} \left[ e^{\frac{\mu f(\zeta) - c(\xi, \zeta)}{\mu \epsilon + \tau}} \right] = \psi^{\tau, \epsilon}(\mu, f, \xi).$$

Indeed, for $(\mu, \mu') \in [\lambda_{\mathrm{up}}^{-1}, \lambda_{\mathrm{low}}^{-1}] \times [\lambda_{\mathrm{up}}^{-1}, \lambda_{\mathrm{low}}^{-1}]$ and $(f, f') \in \mathcal{F} \times \mathcal{F}$, we can write

$$|h(\mu, f) - h(\mu', f')| \leq |h(\mu, f) - h(\mu', f)| + |h(\mu', f) - h(\mu', f')|$$

$$\leq \left( \|\mathcal{F}\|_\infty + \left( \frac{\lambda_{\mathrm{up}} \epsilon}{\epsilon + \lambda_{\mathrm{up}} \tau} \right) \right) |\mu - \mu'| + \lambda_{\mathrm{low}}^{-1} \|f - f'\|_\infty.$$

hence $h$ has Lipschitz constant $\|\mathcal{F}\|_\infty + \lambda_{\mathrm{low}}^{-1} + \left( \frac{\lambda_{\mathrm{up}} \epsilon}{\epsilon + \lambda_{\mathrm{up}} \tau} \right) m_c$. $\qquad \square$

# D   DUAL BOUNDS AND MAXIMAL RADIUS

We establish the existence of a dual lower bound on the true robust risk (Lemma 4.1), for the standard WDRO problem in D.1 and for regularized WDRO in D.2. The results involve the maximal radius introduced in Section 4.2. For the regularized case, an estimate of the dual lower bound is provided.

## D.1   STANDARD WDRO: CONTINUITY AT ZERO OF THE MAXIMAL RADIUS

For $\lambda \geq 0$, we recall the expression of the maximal radius:

$$\rho_{\max}(\lambda) = \inf_{f \in \mathcal{F}} \mathbb{E}_{\xi \sim P}[-\partial_\lambda^+ \phi(\lambda, f, \xi)].$$

**Lemma D.1.** $\rho_{\max}(0) = \rho_{\mathrm{crit}}$ and $\lim_{\lambda \to 0^+} \rho_{\max}(\lambda) = \rho_{\mathrm{crit}}$. In particular, there exists $\lambda_{\mathrm{low}} > 0$ such that $\rho_{\max}(\lambda) \geq \frac{\rho_{\mathrm{crit}}}{4}$ for all $\lambda \in [0, 2\lambda_{\mathrm{low}}]$.

*Proof.* For $\xi \in \Xi$, $f - \lambda c(\xi, \cdot)$ is continuous, hence we can apply the envelope formula (Theorem A.1) and the right-sided derivative of $\phi$ with respect to $\lambda$ is $\partial_\lambda^+ \phi(\lambda, f, \xi) = -\min \{c(\xi, \zeta) : \zeta \in \arg\max_\Xi \{f - \lambda c(\xi, \cdot)\}\}$. For convenience, we use the shorthand

$$c^*(\xi, K) := \min\{c(\xi, z), z \in K\}$$

whenever $K \subset \Xi$ is compact. By integration and taking the infimum over $\mathcal{F}$ we obtain

$$\rho_{\max}(\lambda) = \inf_{f \in \mathcal{F}} \mathbb{E}_{\xi \sim P}[c^*(\xi, \arg\max_\Xi \{f - \lambda c(\xi, \cdot)\})]. \tag{19}$$

In particular, $\rho_{\max}(0) = \rho_{\mathrm{crit}}$.

To prove the result, it is sufficient to show that $\liminf_{k \to \infty} \rho_{\max}(\lambda_k) \geq \rho_{\mathrm{crit}}$ for any positive sequence $(\lambda_k)_{k \in \mathbb{N}}$ converging to 0. Indeed, the functions $\mathbb{E}_{\xi \sim P}[\phi(\cdot, f, \xi)]$ are convex hence their right-sided derivatives $\mathbb{E}_{\xi \sim P}[-\partial_\lambda^+ \phi(\cdot, f, \xi)]$ are nonincreasing, and $\rho_{\max}$ is nonincreasing since it is an infimum over nonincreasing functions. This means $\limsup_{k \to \infty} \rho_{\max}(\lambda_k) \leq \rho_{\max}(0)$ for any sequence $\lambda_k \to 0$.

Now assume toward a contradiction that there exists $\epsilon > 0$ and a sequence $(\lambda_k)_{k \in \mathbb{N}}$ from $\mathbb{R}_+$, such that $\lambda_k \to 0$ as $k \to \infty$, and $\rho_{\max}(\lambda_k) \leq \rho_{\text{crit}} - \epsilon$ for all $k \in \mathbb{N}$. From the expression of $\rho_{\max}$ (19) this means that for each $k$, there exists $f_k$ such that $\mathbb{E}_{\xi \sim P}[c^*(\xi, \arg \max_\Xi f_k - \lambda_k c(\xi, \cdot))] \leq \rho_{\text{crit}} - \frac{\epsilon}{2}$. By compactness of $\mathcal{F}$ with respect to $\|\cdot\|_\infty$, we may assume $(f_k)_{k \in \mathbb{N}}$ to converge to some $f^* \in \mathcal{F}$. In particular, for $\xi \in \Xi$, $f_k, -\lambda_k c(\xi, \cdot)$ converges to $f^*$ as $k \to \infty$.

Let $\xi \in \Xi$ be arbitrary. $(\lambda, f) \mapsto \arg \max_\Xi \{f - \lambda c(\xi, \cdot)\}$ is outer semicontinuous with compact values (Lemma A.2) and $c$ is jointly continuous, hence $(\lambda, f) \mapsto c^*(\xi, \arg \max_\Xi \{f - \lambda c(\xi, \cdot))\}$ is lower semicontinuous, see Lemma A.1. We then have $\liminf_{k \to \infty} c^*(\xi, \arg \max_\Xi \{f_k - \lambda_k c(\xi, \cdot)\}) \geq c^*(\xi, \arg \max_\Xi f^*)$. By integration with respect to $\xi \sim P$, we obtain

$$\mathbb{E}_{\xi \sim P}[c^*(\xi, \arg \max_\Xi f^*)] \leq \mathbb{E}_{\xi \sim P}[\liminf_{k \to \infty} c^*(\xi, \arg \max_\Xi \{f_k - \lambda_k c(\xi, \cdot)\})]$$
$$\leq \liminf_{k \to \infty} \mathbb{E}_{\xi \sim P}[c^*(\xi, \arg \max_\Xi \{f_k - \lambda_k c(\xi, \cdot)\})]$$
$$\leq \rho_{\text{crit}} - \frac{\epsilon}{2}.$$

Since, $\rho_{\text{crit}} \leq \mathbb{E}_{\xi \sim P}[c^*(\xi, \arg \max_\Xi f^*)]$, this yields a contradiction, and allows to conclude. $\square$

## D.2 REGULARIZED WDRO: LIPSCHITZ MAXIMAL RADIUS AND UPPER BOUND

### D.2.1 LIPSCHITZ CONTINUITY OF THE MAXIMAL RADIUS

For $\lambda \geq 0$, we consider the regularized maximal radius,

$$\rho_{\max}^{\tau, \epsilon}(\lambda) = \inf_{f \in \mathcal{F}} \mathbb{E}_{\xi \sim P}[-\partial_\lambda \phi^{\tau, \epsilon}(\lambda, f, \xi)].$$

The following result is the regularized version of Lemma D.1. Compared to the standard setting, the maximal radius is Lipschitz continuous, leading to an estimate of the dual lower bound.

**Lemma D.2.** $\rho_{\max}^{\tau, \epsilon}(0) = \rho_{\text{crit}}^{\tau, \epsilon}$ and $\rho_{\max}^{\tau, \epsilon}$ is Lipschitz continuous on $\mathbb{R}_+$ with constant

$$\frac{2}{\epsilon} \left( \frac{\tau^2}{\epsilon^2} \|\mathcal{F}\|_\infty^2 + m_{2,c} e^{\frac{\|\mathcal{F}\|_\infty}{\epsilon} + \min\left\{ \frac{m_c}{\tau}, \frac{2\|\mathcal{F}\|_\infty m_c}{(\rho - m_c)\epsilon} \right\}} \right).$$

*In particular, setting*

$$\lambda_{\text{low}}^{\tau, \epsilon} := \frac{3\epsilon \rho_{\text{crit}}^{\tau, \epsilon}}{8 \left( \frac{\tau^2}{\epsilon^2} \|\mathcal{F}\|_\infty^2 + m_{2,c} e^{\frac{\|\mathcal{F}\|_\infty}{\epsilon} + \min\left\{ \frac{m_c}{\tau}, \frac{2\|\mathcal{F}\|_\infty m_c}{(\rho - m_c)\epsilon} \right\}} \right)}, \tag{20}$$

*then $\rho_{\max}^{\tau, \epsilon}(\lambda) \geq \frac{\rho_{\text{crit}}^{\tau, \epsilon}}{4}$ for all $\lambda \in [0, 2\lambda_{\text{low}}^{\tau, \epsilon}]$.*

*Proof.* $\phi^{\tau, \epsilon}$ is differentiable with respect to $\lambda$ and we can verify that its derivative is given by

$$\partial_\lambda \phi^{\tau, \epsilon}(\lambda, f, \xi) = -\mathbb{E}_{\zeta \sim \pi_0^{\frac{f - \lambda c(\xi, \cdot)}{\epsilon + \lambda \tau}}(\cdot | \xi)} \left[ \frac{\tau f(\zeta) + \epsilon c(\xi, \zeta)}{\epsilon + \lambda \tau} \right] + \tau \log \mathbb{E}_{\zeta \sim \pi_0(\cdot | \xi)} \left[ e^{\frac{f(\zeta) - \lambda c(\xi, \zeta)}{\epsilon + \lambda \tau}} \right].$$

This gives $\rho_{\max}^{\tau, \epsilon}(0) = \rho_{\text{crit}}^{\tau, \epsilon}$ For $f \in \mathcal{F}$ and $\xi \in \Xi$, our goal is now to compute the Lipschitz constant of $\lambda \mapsto \partial_\lambda \phi^{\tau, \epsilon}(\lambda, f, \xi)$. The Lipschitz constant of $\rho_{\max}^{\tau, \epsilon}$ will then be obtained by integration and taking the infimum over Lipschitz functions. We compute the appropriate quantities:

1. We compute the derivative with respect to $\lambda$ of $u_1 : (\lambda, \zeta) \mapsto - \left( \frac{\tau f(\zeta) + \epsilon c(\xi, \zeta)}{\epsilon + \lambda \tau} \right) e^{\frac{f(\zeta) - \lambda c(\xi, \zeta)}{\epsilon + \lambda \tau}}$:

$$\partial_\lambda u_1(\lambda, \zeta) = \left( \frac{\tau^2 f(\zeta) + \epsilon \tau c(\xi, \zeta)}{(\epsilon + \lambda \tau)^2} + \frac{(\tau f(\zeta) + \epsilon c(\xi, \zeta))^2}{(\epsilon + \lambda \tau)^3} \right) e^{\frac{f(\zeta) - \lambda c(\xi, \zeta)}{\epsilon + \lambda \tau}}.$$

2. We compute the derivative with respect to $\lambda$ of $u_2 : (\lambda, \zeta) \mapsto e^{\frac{f(\zeta) - \lambda c(\xi, \zeta)}{\epsilon + \lambda \tau}}$:

$$\partial_\lambda u_2(\lambda, \zeta) = -\left(\frac{\tau f(\zeta) + \epsilon c(\xi, \zeta)}{(\epsilon + \lambda \tau)^2}\right) e^{\frac{f(\zeta) - \lambda c(\xi, \zeta)}{\epsilon + \lambda \tau}}.$$

3. We compute the derivative of $U_3 : \lambda \mapsto \tau \log \mathbb{E}_{\zeta \sim \pi_0(\cdot | \xi)}\left[e^{\frac{f(\zeta) - \lambda c(\xi, \zeta)}{\epsilon + \lambda \tau}}\right]$:

$$U_3'(\lambda) = -\frac{\mathbb{E}_{\zeta \sim \pi_0(\cdot | \xi)}\left[\left(\frac{\tau^2 f(\zeta) + \tau \epsilon c(\xi, \zeta)}{(\epsilon + \lambda \tau)^2}\right) e^{\frac{f(\zeta) - \lambda c(\xi, \zeta)}{\epsilon + \lambda \tau^2}}\right]}{\mathbb{E}_{\zeta \sim \pi_0(\cdot | \xi)}\left[e^{\frac{f(\zeta) - \lambda c(\xi, \zeta)}{\epsilon + \lambda \tau}}\right]}.$$

Combining 1, 2 and 3, we are able to compute the derivative of $\partial_\lambda \phi^{\tau, \epsilon}$:

$$\partial_\lambda^2 \phi^{\tau, \epsilon}(\lambda, f, \xi) = -\frac{\mathbb{E}_{\zeta \sim \pi_0(\cdot | \xi)}[u_1(\lambda, \zeta)] \mathbb{E}_{\zeta \sim \pi_0(\cdot | \xi)}[\partial_\lambda u_2(\lambda, \zeta)]}{\mathbb{E}_{\zeta \sim \pi_0(\cdot | \xi)}[u_2(\lambda, \zeta)]^2} + U_3'(\lambda)$$

$$= \frac{\mathbb{E}_{\zeta \sim \pi_0(\cdot | \xi)}\left[\frac{(\tau f(\zeta) + \epsilon c(\xi, \zeta))^2}{(\epsilon + \lambda \tau)^3} e^{\frac{f(\zeta) - \lambda c(\xi, \zeta)}{\epsilon + \lambda \tau}}\right]}{\mathbb{E}_{\zeta \sim \pi_0(\cdot | \xi)}\left[e^{\frac{f(\zeta) - \lambda c(\xi, \zeta)}{\epsilon + \lambda \tau}}\right]}$$

$$- \frac{\mathbb{E}_{\zeta \sim \pi_0(\cdot | \xi)}\left[\left(\frac{\tau f(\zeta) + \epsilon c(\xi, \zeta)}{(\epsilon + \lambda \tau)^2}\right) e^{\frac{f(\zeta) - \lambda c(\xi, \zeta)}{\epsilon + \lambda \tau}}\right] \mathbb{E}_{\zeta \sim \pi_0(\cdot | \xi)}\left[\left(\frac{\tau f(\zeta) + \epsilon c(\xi, \zeta)}{\epsilon + \lambda \tau}\right) e^{\frac{f(\zeta) - \lambda c(\xi, \zeta)}{\epsilon + \lambda \tau}}\right]}{\mathbb{E}_{\zeta \sim \pi_0(\cdot | \xi)}\left[e^{\frac{f(\zeta) - \lambda c(\xi, \zeta)}{\epsilon + \lambda \tau}}\right]}$$

$$= \frac{1}{\epsilon + \lambda \tau} \operatorname*{Var}_{\zeta \sim \pi^{\frac{f - \lambda c(\xi, \cdot)}{\epsilon + \lambda \tau}}(\cdot | \xi)}\left(\frac{\tau f(\zeta) + \epsilon c(\xi, \zeta)}{\epsilon + \lambda \tau}\right),$$

where $\operatorname*{Var}_{\zeta \sim \pi^{\frac{f - \lambda c(\xi, \cdot)}{\epsilon + \lambda \tau}}(\cdot | \xi)}$ is the variance with respect to $\pi^{\frac{f - \lambda c(\xi, \cdot)}{\epsilon + \lambda \tau}}(\cdot | \xi)$.

Note that all quantities can be differentiated under the (conditional) expectation since the derivatives with respect to $\lambda$ involve functions that are continuous on the compact sample space $\Xi$ (they are therefore bounded by a constant), see e.g. Theorem A.5.3 from Durrett (2010). By the property of the variance, we obtain

$$|\partial_\lambda^2 \phi^{\tau, \epsilon}(\lambda, f, \xi)| \leq \frac{1}{\epsilon + \lambda \tau} \mathbb{E}_{\zeta \sim \pi_0^{\frac{f - \lambda c(\xi, \cdot)}{\epsilon + \lambda \tau}}(\cdot | \xi)}\left[\left(\frac{\tau f(\zeta) + \epsilon c(\xi, \zeta)}{\epsilon + \lambda \tau}\right)^2\right]$$

$$\leq \frac{2}{\epsilon^3} \mathbb{E}_{\zeta \sim \pi_0^{\frac{f - \lambda c(\xi, \cdot)}{\epsilon + \lambda \tau}}(\cdot | \xi)}\left[\tau^2 \|\mathcal{F}\|_\infty^2 + \epsilon^2 c(\xi, \zeta)^2\right]. \tag{21}$$

Now we bound the right-hand side of the last inequality. First, we have

$$\mathbb{E}_{\zeta \sim \pi_0(\cdot | \xi)}\left[c(\xi, \zeta)^2 e^{\frac{f(\zeta) - \lambda c(\xi, \zeta)}{\epsilon + \lambda \tau}}\right] \leq m_{2,c} e^{\frac{\|\mathcal{F}\|_\infty}{\epsilon}} \tag{22}$$

On the other hand, by Jensen's inequality, we have

$$\mathbb{E}_{\zeta \sim \pi_0(\cdot | \xi)}\left[e^{\frac{f(\zeta) - \lambda c(\xi, \zeta)}{\epsilon + \lambda \tau}}\right] \geq e^{-\frac{\lambda m_c}{\epsilon + \lambda \tau} - \frac{\|\mathcal{F}\|_\infty}{\epsilon}} \tag{23}$$

We have the alternatives $\frac{\lambda m_c}{\epsilon + \lambda \tau} \leq \frac{\lambda_{\mathrm{up}} m_c}{\epsilon} = \frac{2\|\mathcal{F}\|_\infty m_c}{(\rho - m_c)\epsilon}$ in any case, and $\frac{\lambda m_c}{\epsilon + \lambda \tau} \leq \frac{m_c}{\tau}$ whenever $\tau > 0$. This means $\frac{\lambda m_c}{\epsilon + \lambda \tau} \leq \min\left\{\frac{m_c}{\tau}, \frac{2\|\mathcal{F}\|_\infty m_c}{(\rho - m_c)\epsilon}\right\}$.

Dividing (22) by (23), we obtain $\mathbb{E}_{\zeta \sim \pi_0^{\frac{f - \lambda c(\xi, \cdot)}{\epsilon + \lambda \tau}}(\cdot | \xi)}\left[c(\xi, \zeta)^2\right] \leq m_{2,c} e^{\min\left\{\frac{m_c}{\tau}, \frac{2\|\mathcal{F}\|_\infty m_c}{(\rho - m_c)\epsilon}\right\}} e^{\frac{2\|\mathcal{F}\|_\infty}{\epsilon}}$.
Reinjecting this inequality in (21) gives

$$|\partial_\lambda^2 \phi^{\tau,\epsilon}(\lambda, f, \xi)| \leq \frac{2}{\epsilon}\left(\frac{\tau^2}{\epsilon^2}\|\mathcal{F}\|_\infty^2 + m_{2,c}e^{\frac{2\|\mathcal{F}\|_\infty}{\epsilon} + \min\left\{\frac{m_c}{\tau}, \frac{2\|\mathcal{F}\|_\infty m_c}{(\rho - m_c)\epsilon}\right\}}\right) := L. \tag{24}$$

This means that for $f \in \mathcal{F}$, the function $g : (\lambda, f) \mapsto \mathbb{E}_{\xi \sim P}[-\partial_\lambda \phi^{\tau,\epsilon}(\lambda, f, \xi)]$ is $L$-Lipschitz where $L$ is given by (24).

We then show that $\rho_{\max}^{\tau,\epsilon} := \inf_{f \in \mathcal{F}} g(\cdot, f)$ is $L$-Lipschitz continuous. Let $(\lambda, \lambda') \in \mathbb{R}^2$, and let $(f_k)_{k \in \mathbb{N}}$ be a sequence from $\mathcal{F}$ such that $g(\lambda', f_k) \underset{k \to \infty}{\to} \rho_{\max}^{\tau,\epsilon}(\lambda')$. Then by definition of $\rho_{\max}^{\tau,\epsilon}$, we have for all $k \in \mathbb{N}$,

$$\rho_{\max}^{\tau,\epsilon}(\lambda) - g(\lambda', f_k) \leq g(\lambda, f_k) - g(\lambda', f_k) \leq L|\lambda - \lambda'|.$$

Taking the limit as $k \to \infty$ gives $\rho_{\max}^{\tau,\epsilon}(\lambda) - \rho_{\max}^{\tau,\epsilon}(\lambda') \leq L|\lambda - \lambda'|$. Exchanging the roles of $\lambda$ and $\lambda'$ gives $|\rho_{\max}^{\tau,\epsilon}(\lambda) - \rho_{\max}^{\tau,\epsilon}(\lambda')| \leq L|\lambda - \lambda'|$, hence $\rho_{\max}^{\tau,\epsilon}$ is $L$-Lipschitz.

Now, set $2\lambda_{\text{low}}^{\tau,\epsilon} := \sup\{\lambda \in \mathbb{R}_+ : \rho_{\max}^{\tau,\epsilon}(\lambda) \geq \rho_{\text{crit}}^{\tau,\epsilon}/4\}$. Then either $\lambda_{\text{low}}^{\tau,\epsilon} = \infty$ (in which case any value $\lambda_{\text{low}}^{\tau,\epsilon}$ satisfies the desired property), or by continuity of $\rho_{\max}^{\tau,\epsilon}$, $\rho_{\max}^{\tau,\epsilon}(2\lambda_{\text{low}}^{\tau,\epsilon}) = \rho_{\text{crit}}^{\tau,\epsilon}/4$ and we have $\rho_{\text{crit}}^{\tau,\epsilon} - 2L\lambda_{\text{low}}^{\tau,\epsilon} \leq \rho_{\max}(2\lambda_{\text{low}}^{\tau,\epsilon}) = \rho_{\text{crit}}^{\tau,\epsilon}/4$. Finally, we obtain (20). □

### D.2.2 DUAL UPPER BOUND

The following result allows to bound the dual solution above. This step is specific to the regularized setting, see in particular Proposition F.4 which illustrates this requirement.

**Lemma D.3** (Upper bound for the regularized problem Lemma 4.3). *Assume $\rho > m_c$ and let $\lambda_{\text{up}} := \frac{2\|\mathcal{F}\|_\infty}{\rho - m_c}$. For all $f \in \mathcal{F}$ and $Q \in \mathcal{P}(\Xi)$,*

$$\inf_{\lambda \in [0, \infty)} \{\lambda\rho + \mathbb{E}_{\xi \sim Q}[\phi^{\tau,\epsilon}(\lambda, f, \xi)]\} = \inf_{\lambda \in [0, \lambda_{\text{up}}]} \{\lambda\rho + \mathbb{E}_{\xi \sim Q}[\phi^{\tau,\epsilon}(\lambda, f, \xi)]\}.$$

*Proof.* Let $\xi \in \Xi$ be arbitrary. Recall that

$$\partial_\lambda \phi^{\tau,\epsilon}(\lambda, f, \xi) = -\mathbb{E}_{\zeta \sim \pi_0^{\frac{f - \lambda c(\xi, \cdot)}{\epsilon + \lambda \tau}}(\cdot | \xi)}\left[\frac{\tau f(\zeta) + \epsilon c(\xi, \zeta)}{\epsilon + \lambda\tau}\right] + \tau \log \mathbb{E}_{\zeta \sim \pi_0(\cdot | \xi)}\left[e^{\frac{f(\zeta) - \lambda c(\xi, \zeta)}{\epsilon + \lambda\tau}}\right].$$

We bound $-\partial_\lambda \phi^{\tau,\epsilon}(\lambda, f, \xi)$ above, uniformly in $f \in \mathcal{F}$ and $\xi \in \Xi$. For readability of the proof, we set $\tilde{\pi}_0 = \pi_0^{\frac{f - \lambda c(\xi, \cdot)}{\epsilon + \lambda\tau}}$ with a slight abuse of notation. In this case, we have

$$\mathbb{E}_{\zeta \sim \tilde{\pi}_0(\cdot|\xi)}\left[\frac{\tau f(\zeta) + \epsilon c(\xi, \zeta)}{\epsilon + \lambda\tau}\right] = \mathbb{E}_{\zeta \sim \tilde{\pi}_0(\cdot|\xi)}\left[\frac{\lambda\tau f(\zeta) + \lambda\epsilon c(\xi, \zeta) - \epsilon f(\zeta) + \epsilon f(\zeta)}{\lambda(\epsilon + \lambda\tau)}\right]$$

$$= \frac{1}{\lambda}\mathbb{E}_{\zeta \sim \tilde{\pi}_0(\cdot|\xi)}[f(\zeta)] - \frac{\epsilon}{\lambda}\mathbb{E}_{\zeta \sim \tilde{\pi}_0(\cdot|\xi)}\left[\frac{f(\zeta) - \lambda c(\xi, \zeta)}{\epsilon + \lambda\tau}\right]$$

$$\leq \frac{\|\mathcal{F}\|_\infty}{\lambda} - \frac{\epsilon}{\lambda}\log \mathbb{E}_{\zeta \sim \pi_0(\cdot|\xi)}\left[e^{\frac{f(\zeta) - \lambda c(\xi, \zeta)}{\epsilon + \lambda\tau}}\right]$$

$$\leq \frac{\|\mathcal{F}\|_\infty}{\lambda} - \frac{\epsilon}{\lambda(\epsilon + \lambda\tau)}\left(\mathbb{E}_{\zeta \sim \pi_0(\cdot|\xi)}[f(\zeta) - \lambda c(\xi, \zeta)]\right)$$

$$\leq \frac{\|\mathcal{F}\|_\infty}{\lambda} + \frac{\epsilon\|\mathcal{F}\|_\infty}{\lambda(\epsilon + \lambda\tau)} + \frac{\epsilon m_c}{\epsilon + \lambda\tau}, \tag{25}$$

where for the third line, we used Lemma C.2, and for the fourth line, we used Jensen's inequality. On the other hand,

$$-\tau\log\mathbb{E}_{\zeta \sim \pi_0(\cdot|\xi)}\left[e^{\frac{f(\zeta) - \lambda c(\xi, \zeta)}{\epsilon + \lambda\tau}}\right] \leq -\frac{\tau}{\epsilon + \lambda\tau}\mathbb{E}_{\zeta \sim \pi_0(\cdot|\xi)}[f(\zeta) - \lambda c(\xi, \zeta)]$$

$$\leq \frac{\lambda\tau}{\lambda(\epsilon + \lambda\tau)}\|\mathcal{F}\|_\infty + \frac{\lambda\tau}{\epsilon + \lambda\tau}m_c \tag{26}$$

Summing (25) and (26) gives

$$-\partial_\lambda \phi^{\tau,\epsilon}(\lambda, f, \xi) \leq \frac{2\|\mathcal{F}\|_\infty}{\lambda} + m_c,$$

whence assuming $\rho > m_c$, and taking $\lambda = \lambda_{\text{up}} := \frac{2\|\mathcal{F}\|_\infty}{\rho - m_c}$, we obtain for all $f \in \mathcal{F}$ and all $\xi \in \Xi$,

$$0 \leq \rho + \partial_\lambda \phi^{\tau,\epsilon}(\lambda_{\text{up}}, f, \xi).$$

Integrating with respect to a distribution $Q \in \mathcal{P}(\Xi)$ yields

$$0 \leq \rho + \mathbb{E}_{\xi \sim Q}[\partial_\lambda \phi^{\tau,\epsilon}(\lambda_{\text{up}}, f, \xi)],$$

which is the derivative at $\lambda_{\text{up}}$ of the convex function $\lambda \mapsto \lambda\rho + \mathbb{E}_{\xi \sim Q}[\phi^{\tau,\epsilon}(\lambda, f, \xi)]$. This means

$$\inf_{\lambda \in [0,\infty)} \{\lambda\rho + \mathbb{E}_{\xi \sim Q}[\phi^{\tau,\epsilon}(\lambda, f, \xi)]\} = \inf_{\lambda \in [0,\lambda_{\text{up}})} \left\{\lambda\rho + \mathbb{E}_{\xi \sim \widehat{P}_n}[\phi^{\tau,\epsilon}(\lambda, f, \xi)]\right\}.$$

$\square$

## E PROOF OF THE MAIN RESULTS

In this section, we prove the main results of the paper. First, we establish the core concentration results in E.1 that apply to standard and regularized WDRO. In particular, we establish the dual lower bound with high probability on the empirical robust risk. We deduce Theorems 3.1 and 3.2 in E.2 by computing the generalization constants. In E.3 we obtain the excess bounds (Proposition 3.1 and Proposition 3.3). Finally, the results on linear models (Proposition 3.2) are found in E.4.

### E.1 DUAL BOUNDS WITH HIGH PROBABILITY ON THE EMPIRICAL PROBLEM

All the results of this subsection hold for both standard and regularized cases. The proofs hold *as is*, replacing $\phi$, $\psi$, $\rho_{\text{crit}}$, $\rho_{\text{max}}$ and $\lambda_{\text{low}}$ by $\phi^{\tau,\epsilon}$, $\psi^{\tau,\epsilon}$, $\rho_{\text{crit}}^{\tau,\epsilon}$, $\rho_{\text{max}}^{\tau,\epsilon}$ and $\lambda_{\text{low}}^{\tau,\epsilon}$ respectively.

For $\lambda \geq 0$, we recall the expression of the maximal radius:

$$\rho_{\text{max}}(\lambda) = \inf_{f \in \mathcal{F}} \mathbb{E}_{\xi \sim P}[-\partial_\lambda^+ \phi(\lambda, f, \xi)].$$

**Problem's constants.** Before proving the next results, we introduce several quantities:

**Proposition E.1** (Dual lower bound in the true problem). *Under Assumption 2.1, there exists $\lambda_{\text{low}} > 0$ such that for all $\lambda \in [0, 2\lambda_{\text{low}}]$, $\rho_{\text{max}}(\lambda) \geq \frac{\rho_{\text{crit}}}{4}$. In particular, $\mathbb{E}_{\xi \sim P}[\partial_\lambda^+ \phi(\lambda, f, \xi)] \leq -\frac{\rho_{\text{crit}}}{4}$ for all $f \in \mathcal{F}$.*

*Proof.* This comes from $\lim_{\lambda \to 0^+} \rho_{\text{max}}(\lambda) = \rho_{\text{crit}}$. See lemma D.1 for standard WDRO and lemma D.2 for the regularized case. $\square$

**Remark E.1** (Refining the degeneracy threshold). *The constant $\lambda_{\text{low}}$ may be refined to fix another threshold than $\frac{\rho_{\text{crit}}}{4}$. More precisely, for any $\eta \in (0, 1)$, we may also find $\lambda_{\text{low}} > 0$ such that for all $\lambda \in [0, 2\lambda_{\text{low}}]$, $\rho_{\text{max}}(\lambda) \geq \eta\rho_{\text{crit}}$. We choose $\eta = \frac{1}{4}$ in Proposition E.1 to be consistent with the study of linear models from Section 3.2.*

For the next results, we define the following quantities:

- $\Phi$ is the length of a segment $I$ such that $\phi(\lambda, f, \xi) \in I$ for all $\lambda \in \{\lambda_{\text{low}}, 2\lambda_{\text{low}}\}$, $f \in \mathcal{F}$ and $\xi \in \Xi$.

- $\Psi$ is the length of a segment $J$ such that $\psi(\mu, f, \xi) \in J$ for all $\mu \in (0, \lambda_{\text{low}}^{-1}]$, $f \in \mathcal{F}$ and $\xi \in \Xi$.

- $L_\psi$ and $\lambda_{\text{up}} \in [0, \infty]$ are such that $\psi(\cdot, \cdot, \xi)$ is $L_\psi$-Lipschitz on $[\lambda_{\text{up}}^{-1}, \lambda_{\text{low}}^{-1}] \times \mathcal{F}$ for all $\xi \in \Xi$.

Let $\lambda_{\text{low}} > 0$ be the dual lower bound given by Proposition E.1. We now to show this quantity is a lower bound on the empirical robust risk:

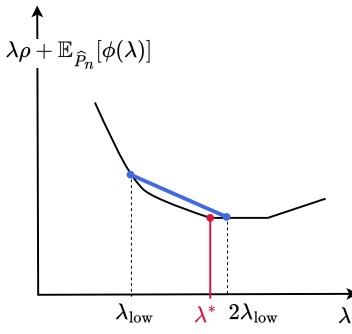

Figure 2: Bounding from below the empirical dual solution $\lambda^*$ expresses as a slope condition (thanks to convexity of the objective).

**Proposition E.2** (Dual lower bound with high probability). *Under Assumption 2.1, let $\lambda_{\text{low}}$ be given by Proposition E.1, and $\lambda_{\text{up}} \in [\lambda_{\text{low}}, \infty]$. If $\rho \leq \frac{\rho_{\text{crit}}}{4} - \frac{C(\delta)}{\sqrt{n}}$ where $C(\delta) := \frac{96\mathcal{I}(\mathcal{F}, \|\cdot\|_\infty)}{\lambda_{\text{low}}} + \frac{2\Phi}{\lambda_{\text{low}}}\sqrt{2\log\frac{4}{\delta}}$, then with probability $1 - \frac{\delta}{2}$, for all $f \in \mathcal{F}$,*

$$\inf_{\lambda \in [0, \lambda_{\text{up}})} \left\{ \lambda\rho + \mathbb{E}_{\xi \sim \widehat{P}_n}[\phi(, \lambda, f, \xi)] \right\} = \inf_{\lambda \in [\lambda_{\text{low}}, \lambda_{\text{up}})} \left\{ \lambda\rho + \mathbb{E}_{\xi \sim \widehat{P}_n}[\phi(\lambda, f, \xi)] \right\}.$$

*Proof.* The proof consists in using the convexity of $\phi(\cdot, f, \xi)$. Indeed, given a convex function $g$ over $\mathbb{R}^+$, the infimum of $g$ has to occur on an interval $[\lambda_{\text{low}}, +\infty]$ if $g$ has a negative slope between $\lambda_{\text{low}}$ and $2\lambda_{\text{low}}$ (Figure 2):

$$\frac{g(2\lambda_{\text{low}}) - g(\lambda_{\text{low}})}{\lambda_{\text{low}}} \leq 0 \implies \inf_{\lambda \geq \lambda_{\text{low}}} g(\lambda) = \inf_{\lambda \geq 0} g(\lambda).$$

We want this condition satisfied for the empirical Lagrangian function $g(\lambda) = \lambda\rho + \mathbb{E}_{\widehat{P}_n}[\phi(\lambda, f)]$ with high probability. For convenience, this can be expressed with the slope of $\mathbb{E}_{\widehat{P}_n}[\phi(\cdot, f)]$:

$$\widehat{s}(f) := \frac{\mathbb{E}_{\widehat{P}_n}[\phi(2\lambda_{\text{low}}, f)] - \mathbb{E}_{\widehat{P}_n}[\phi(\lambda_{\text{low}}, f)]}{\lambda_{\text{low}}} \leq -\rho. \tag{27}$$

This is the condition we aim to obtain. To this end, we proceed by comparing the empirical slope to the true one, that is $s(f) := \frac{\mathbb{E}_P[\phi(2\lambda_{\text{low}}, f)] - \mathbb{E}_P[\phi(\lambda_{\text{low}}, f)]}{\lambda_{\text{low}}}$. We can show that any function $(f, \xi) \mapsto \phi(\lambda, f, \xi)$, with $\lambda \in \mathbb{R}_+$, satisfies the requirements for the concentration theorem Theorem A.2, which is done in Lemma C.1 and Lemma C.3. Consequently, we can apply the concentration theorem twice, on each function $\phi(2\lambda_{\text{low}}, \cdot, \cdot)$ and $\phi(2\lambda_{\text{low}}, \cdot, \cdot)$, to obtain that $\widehat{s}(f)$ approximates $s(f)$ with probability at least $1 - \frac{\delta}{2}$,

$$\forall f \in \mathcal{F}, \quad \widehat{s}(f) \leq s(f) + \frac{\beta}{\sqrt{n}},$$

where $C(\delta) > 0$ will be computed afterwards. On the other hand, $s(f) \leq \mathbb{E}_P[\partial_\lambda^+ \phi(2\lambda_{\text{low}}, f)]$ by convexity of $\phi$, hence $s(f) \leq -\frac{\rho_{\text{crit}}}{4}$ by Proposition E.1. This means $\widehat{s}(f) \leq \frac{\beta}{\sqrt{n}} - \frac{\rho_{\text{crit}}}{4}$ hence we have the desired condition (27) when $\rho \leq \frac{\rho_{\text{crit}}}{4} - \frac{C(\delta)}{\sqrt{n}}$, and with probability at least $1 - \frac{\delta}{2}$, for all $f \in \mathcal{F}$,

$$\inf_{\lambda \in [0, \lambda_{\text{up}})} \left\{ \lambda\rho + \mathbb{E}_{\xi \sim \widehat{P}_n}[\phi(\lambda, f, \xi)] \right\} = \inf_{\lambda \in [\lambda_{\text{low}}, \lambda_{\text{up}})} \left\{ \lambda\rho + \mathbb{E}_{\xi \sim \widehat{P}_n}[\phi(\lambda, f, \xi)] \right\}.$$

*Concentration constant $C(\delta)$.* We now compute $C(\delta)$. Let $\lambda \in \{\lambda_{\text{low}}, 2\lambda_{\text{low}}\}$. By Theorem A.2, we have with probability at least $1 - \frac{\delta}{4}$, for all $f \in \mathcal{F}$,

$$\mathbb{E}_{\xi \sim \widehat{P}_n}[\phi(2\lambda_{\text{low}}, f, \xi)] - \mathbb{E}_{\xi \sim P}[\phi(2\lambda_{\text{low}}, f, \xi)] \leq \frac{48\mathcal{I}(\mathcal{F}, \|\cdot\|_\infty)}{\sqrt{n}} + \Phi\sqrt{\frac{2\log\frac{4}{\delta}}{n}} \tag{28}$$

and with probability at least $1 - \frac{\delta}{4}$, for all $f \in \mathcal{F}$,

$$\mathbb{E}_{\xi \sim P}[\phi(\lambda_{\text{low}}, f, \xi)] - \mathbb{E}_{\xi \sim \widehat{P}_n}[\phi(\lambda_{\text{low}}, f, \xi)] \leq \frac{48\mathcal{I}(\mathcal{F}, \|\cdot\|_\infty)}{\sqrt{n}} + \Phi\sqrt{\frac{2\log\frac{4}{\delta}}{n}}. \tag{29}$$

We set $C'(\delta) := 48\mathcal{I}(\mathcal{F}, \|\cdot\|_\infty) + \Phi\sqrt{2\log\frac{4}{\delta}}$. Intersecting the events (28) and (29), we obtain with probability $1 - \frac{\delta}{2}$, for all $f \in \mathcal{F}$,

$$\frac{\mathbb{E}_{\xi \sim \widehat{P}_n}[\phi(2\lambda_{\text{low}}, f, \xi)] - \mathbb{E}_{\xi \sim \widehat{P}_n}[\phi(\lambda_{\text{low}}, f, \xi)]}{\lambda_{\text{low}}}$$

$$\leq \frac{1}{\lambda_{\text{low}}} \left( \mathbb{E}_{\xi \sim P}[\phi(2\lambda_{\text{low}}, f, \xi)] - \mathbb{E}_{\xi \sim P}[\phi(\lambda_{\text{low}}, f, \xi)] + \frac{2C'(\delta)}{\sqrt{n}} \right)$$

$$\leq \mathbb{E}_{\xi \sim P}[\partial_\lambda^+ \phi(2\lambda_{\text{low}}, f, \xi)] + \frac{2C'(\delta)}{\lambda_{\text{low}}\sqrt{n}}$$

$$\leq -\frac{\rho_{\text{crit}}}{4} + \frac{2C'(\delta)}{\lambda_{\text{low}}\sqrt{n}}, \tag{30}$$

where we recall that for $\lambda_{\text{low}} > 0$, satisfies for all $\lambda \in [0, 2\lambda_{\text{low}}]$ and all $f \in \mathcal{F}$, $\mathbb{E}_{\xi \sim P}[\partial^+ \phi(\lambda, f, \xi)] \leq -\frac{\rho_{\text{crit}}}{4}$. This means $C(\delta) = 2C'(\delta)/\lambda_{\text{low}}$ and we have the desired expression.

$\square$

This implies a generalization bound on the dual problem of (regularized) WDRO:

**Proposition E.3** (Generalization bound on the dual problem). *Under Assumption 2.1, let $\lambda_{\text{low}} > 0$ be given by Proposition E.1. If $\frac{B(\delta)}{\sqrt{n}} \leq \rho \leq \frac{\rho_{\text{crit}}}{4} - \frac{C(\delta)}{\sqrt{n}}$ where*

- $B(\delta) = 48 L_\psi \left( \mathcal{I}(\mathcal{F}, \|\cdot\|_\infty) + \frac{2}{\lambda_{\text{low}}} \right) + \Psi \sqrt{2 \log \frac{2}{\delta}}$,

- $C(\delta) = \frac{96 \mathcal{I}(\mathcal{F}, \|\cdot\|_\infty)}{\lambda_{\text{low}}} + \frac{2\Phi}{\lambda_{\text{low}}} \sqrt{2 \log \frac{4}{\delta}}$,

*then with probability at least $1 - \delta$, for all $f \in \mathcal{F}$,*

$$\inf_{\lambda \in [0, \lambda_{\text{up}})} \left\{ \lambda \rho + \mathbb{E}_{\xi \sim \widehat{P}_n}[\phi(\lambda, f, \xi)] \right\} \geq \inf_{\lambda \in [0, \infty)} \left\{ \lambda \left( \rho - \frac{B(\delta)}{\sqrt{n}} \right) + \mathbb{E}_{\xi \sim P}[\phi(\lambda, f, \xi)] \right\}.$$

*Proof.* We assume $\lambda_{\text{up}} > \lambda_{\text{low}}$. By Theorem A.2, applied to $(\mu, f) \mapsto \mu \phi(\mu^{-1}, f, \xi)$, we obtain with probability at least $1 - \frac{\delta}{2}$,

$$\alpha_n := \sup_{(\mu, f) \in (\lambda_{\text{up}}^{-1}, \lambda_{\text{low}}^{-1}] \times \mathcal{F}} \left\{ \mathbb{E}_{\xi \sim P}[\psi(\mu, f, \xi)] - \mathbb{E}_{\xi \sim \widehat{P}_n}[\psi(\mu, f, \xi)] \right\} \leq \frac{B(\delta)}{\sqrt{n}} \tag{31}$$

where $B(\delta) = 48 L_\psi \mathcal{I}([0, \lambda_{\text{low}}^{-1}] \times \mathcal{F}, \text{dist}) + \Psi \sqrt{2 \log \frac{2}{\delta}}$ and $\text{dist}((\mu, f), (\mu', f')) := |\mu - \mu'| + \|f - f'\|_\infty$. Furthermore, we have the inequality

$$\mathcal{I}([0, \lambda_{\text{low}}^{-1}] \times \mathcal{F}) \leq \mathcal{I}(\mathcal{F}, \|\cdot\|_\infty) + \frac{1}{2\lambda_{\text{low}}}(1 + 2\log 2) \leq \mathcal{I}(\mathcal{F}, \|\cdot\|_\infty) + \frac{2}{\lambda_{\text{low}}},$$

see Lemma A.3, hence we may refine $B(\delta)$ as $B(\delta) = 48 L_\psi \left( \mathcal{I}(\mathcal{F}, \|\cdot\|_\infty) + \frac{2}{\lambda_{\text{low}}} \right) + \Psi \sqrt{2 \log \frac{2}{\delta}}$.

By Proposition E.2, if $\rho \leq \frac{\rho_{\text{crit}}}{4} - \frac{C(\delta)}{\sqrt{n}}$, then with probability at least $1 - \frac{\delta}{2}$, for all $f \in \mathcal{F}$,

$$\inf_{\lambda \in [0, \lambda_{\text{up}})} \left\{ \lambda \rho + \mathbb{E}_{\xi \sim \widehat{P}_n}[\phi(\lambda, f, \xi)] \right\} = \inf_{\lambda \in [\lambda_{\text{low}}, \lambda_{\text{up}})} \left\{ \lambda \rho + \mathbb{E}_{\xi \sim \widehat{P}_n}[\phi(\lambda, f, \xi)] \right\}. \tag{32}$$

Finally, combining (32) and (31), and if

$$\frac{B(\delta)}{\sqrt{n}} \leq \rho \leq \frac{\rho_{\text{crit}}}{4} - \frac{C(\delta)}{\sqrt{n}},$$

we can write with probability $1 - \delta$, for all $f \in \mathcal{F}$,

$$\inf_{\lambda \in [0, \lambda_{\text{up}})} \left\{ \lambda \rho + \mathbb{E}_{\xi \sim \widehat{P}_n}[\phi(\lambda, f, \xi)] \right\} = \inf_{\lambda \in [\lambda_{\text{low}}, \lambda_{\text{up}})} \left\{ \lambda \rho + \mathbb{E}_{\xi \sim \widehat{P}_n}[\phi(\lambda, f, \xi)] \right\}$$

$$\geq \inf_{\lambda \in [\lambda_{\text{low}}, \lambda_{\text{up}})} \left\{ \lambda \rho + \mathbb{E}_{\xi \sim P}[\phi(\lambda, f, \xi)] - \lambda \frac{\mathbb{E}_{\xi \sim P}[\phi(\lambda, f, \xi)] - \mathbb{E}_{\xi \sim \widehat{P}_n}[\phi(\lambda, f, \xi)]}{\lambda} \right\}$$

$$\geq \inf_{\lambda \in [\lambda_{\text{low}}, \lambda_{\text{up}})} \left\{ \lambda \rho + \mathbb{E}_{\xi \sim P}[\phi(\lambda, f, \xi)] - \lambda \alpha_n \right\}$$

$$\geq \inf_{\lambda \in [\lambda_{\text{low}}, \lambda_{\text{up}})} \left\{ \lambda \left( \rho - \frac{B(\delta)}{\sqrt{n}} \right) + \mathbb{E}_{\xi \sim P}[\phi(\lambda, f, \xi)] \right\}$$

$$\geq \inf_{\lambda \in [0, \infty)} \left\{ \lambda \left( \rho - \frac{B(\delta)}{\sqrt{n}} \right) + \mathbb{E}_{\xi \sim P}[\phi(\lambda, f, \xi)] \right\},$$

If $\lambda_{\mathrm{up}} \leq \lambda_{\mathrm{low}}$, this means, by convexity of the inner function,

$$\inf_{\lambda \in [0, \lambda_{\mathrm{up}})} \left\{ \lambda \rho + \mathbb{E}_{\xi \sim \widehat{P}_n} [\phi(\lambda, f, \xi)] \right\} = \lambda_{\mathrm{low}} \rho + \mathbb{E}_{\xi \sim \widehat{P}_n} [\phi(\lambda_{\mathrm{low}}, f, \xi)]$$

$$\geq \lambda_{\mathrm{low}} (\rho - \alpha'_n) + \mathbb{E}_{\xi \sim P} [\phi(\lambda_{\mathrm{low}}, f, \xi)]$$

$$\geq \inf_{\lambda \in [0, \infty)} \left\{ \lambda \left( \rho - \frac{B(\delta)}{\sqrt{n}} \right) + \mathbb{E}_{\xi \sim P} [\phi(\lambda, f, \xi)] \right\},$$

where we refined $\alpha_n$ into $\alpha'_n = \sup_{f \in \mathcal{F}} \left\{ \mathbb{E}_{\xi \sim P} [\psi(\lambda_{\mathrm{low}}^{-1}, f, \xi)] - \mathbb{E}_{\xi \sim \widehat{P}_n} [\psi(\lambda_{\mathrm{low}}^{-1}, f, \xi)] \right\}$. $\qquad \square$

### E.2 GENERALIZATION BOUNDS

We are now ready to prove the generalization bounds. The following is an extended version of the generalization result in standard WDRO (Theorem 3.1). Note that the extended bound (33) involves a control of $R_{\rho - \frac{\alpha}{\sqrt{n}}}(f)$, which means that $\widehat{R}_\rho(f)$ also generalizes well against distribution shifts.

**Theorem E.1** (Generalization guarantee, standard WDRO). *Under Assumption 2.1, there exists $\lambda_{\mathrm{low}} > 0$ such that if*

$$\frac{\alpha}{\sqrt{n}} < \rho \leq \frac{\rho_{\mathrm{crit}}}{4} - \frac{\beta}{\sqrt{n}},$$

*where*

- $\alpha = 48 \left( \|\mathcal{F}\|_\infty + \frac{1}{\lambda_{\mathrm{low}}} \right) \left( \mathcal{I}(\mathcal{F}, \|\cdot\|_\infty) + \frac{2}{\lambda_{\mathrm{low}}} \right) + \frac{2\|\mathcal{F}\|_\infty}{\lambda_{\mathrm{low}}} \sqrt{2 \log \frac{2}{\delta}}$

- $\beta = \frac{96 \mathcal{I}(\mathcal{F}, \|\cdot\|_\infty)}{\lambda_{\mathrm{low}}} + \frac{4\|\mathcal{F}\|_\infty}{\lambda_{\mathrm{low}}} \sqrt{2 \log \frac{4}{\delta}},$

*then with probability at least $1 - \delta$, for all $f \in \mathcal{F}$,*

$$\widehat{R}_\rho(f) \geq R_{\rho - \frac{\alpha}{\sqrt{n}}}(f) \geq \mathbb{E}_{\xi \sim P} [f(\xi)]. \tag{33}$$

*In particular, for any $\rho > \frac{\alpha}{\sqrt{n}}$ and $n > 16(\alpha + \beta)^2 / \rho_{\mathrm{crit}}^2$, with probability at least $1 - \delta$, $\widehat{R}_\rho(f) \geq \mathbb{E}_{\xi \sim P} [f(\xi)]$ for all $f \in \mathcal{F}$.*

*Proof.* Under Assumption 2.1, let $\lambda_{\mathrm{low}}$ be given by Proposition E.2. Our goal is to apply Proposition E.3 in the standard WDRO case and to compute its constants thanks to Lemma C.1. By Lemma C.1, we have the following constants:

- $\Phi = 2\|\mathcal{F}\|_\infty$,

- $\Psi = \frac{2\|\mathcal{F}\|_\infty}{\lambda_{\mathrm{low}}}$,

- $\lambda_{\mathrm{up}} = \infty$, and $L_\psi = \|\mathcal{F}\|_\infty + \lambda_{\mathrm{low}}^{-1}$.

$\alpha$ corresponds to $B(\delta)$ in Proposition E.3 and $\beta$ corresponds $C(\delta)$, whence we obtain the desired expressions for $\alpha$ and $\beta$ with the quantities above.

By strong duality, Proposition B.1, $R_\varrho(f)$ and $\widehat{R}_\varrho(f)$ admit the representations

$$R_\varrho(f) = \inf_{\lambda \in [0, \infty)} \left\{ \lambda \varrho + \mathbb{E}_{\xi \sim P} [\phi(\lambda, f, \xi)] \right\}$$

$$\widehat{R}_\varrho(f) = \inf_{\lambda \in [0, \infty)} \left\{ \lambda \varrho + \mathbb{E}_{\xi \sim \widehat{P}_n} [\phi(\lambda, f, \xi)] \right\},$$

for any $\varrho \geq 0$ and $f \in \mathcal{F}$. By Proposition E.3, if $\frac{\alpha}{\sqrt{n}} < \rho \leq \frac{\rho_{\mathrm{crit}}}{4} - \frac{\beta}{\sqrt{n}}$, then with probability at least $1 - \delta$, we have for all $f \in \mathcal{F}$, $\widehat{R}_\rho(f) \geq R_{\rho - \frac{\alpha}{\sqrt{n}}}(f)$, hence the first part of the result.

As to the last statement, if $n > 16(\alpha + \beta)^2/\rho_{\mathrm{crit}}^2$, $\frac{\alpha}{\sqrt{n}} < \frac{\rho_{\mathrm{crit}}}{4} - \frac{\beta}{\sqrt{n}}$. For any $\frac{\alpha}{\sqrt{n}} < \rho \le \frac{\rho_{\mathrm{crit}}}{4} - \frac{\beta}{\sqrt{n}}$, with probability at least $1 - \delta$, $\widehat{R}_\rho(f) \ge \mathbb{E}_{\xi \sim P}[f(\xi)]$ for all $f \in \mathcal{F}$ as shown previously. For $\rho \ge \frac{\rho_{\mathrm{crit}}}{4} - \frac{\beta}{\sqrt{n}}$, since the quantity $\widehat{R}_\rho(f)$ is non-decreasing with respect to $\rho$, we also have $\widehat{R}_\rho(f) \ge \mathbb{E}_{\xi \sim P}[f(\xi)]$. $\qquad \square$

The next result corresponds to the generalization guarantee for WDRO with double regularization (Theorem 3.2).

**Theorem E.2** (Generalization guarantee, regularized WDRO). *Under Assumption 2.1, there exists* $\lambda_{\mathrm{low}} > 0$ *such that if*

$$\max\left\{m_c, \frac{\alpha^{\tau,\epsilon}}{\sqrt{n}}\right\} < \rho \le \frac{\rho_{\mathrm{crit}}^{\tau,\epsilon}}{4} - \frac{\beta^{\tau,\epsilon}}{\sqrt{n}},$$

*where* $\alpha^{\tau,\epsilon}$ *and* $\beta^{\tau,\epsilon}$ *are the two constants*

- $\alpha^{\tau,\epsilon} = 48\left(\|\mathcal{F}\|_\infty + \frac{1}{\lambda_{\mathrm{low}}^{\tau,\epsilon}} + \frac{2\|\mathcal{F}\|_\infty m_c \epsilon}{\epsilon(\rho - m_c) + 2\tau\|\mathcal{F}\|_\infty}\right)\left(\mathcal{I}(\mathcal{F}, \|\cdot\|_\infty) + \frac{2}{\lambda_{\mathrm{low}}^{\tau,\epsilon}}\right)$
  $\quad + \left(\frac{2\|\mathcal{F}\|_\infty}{\lambda_{\mathrm{low}}^{\tau,\epsilon}} + m_c\right)\sqrt{2\log\frac{2}{\delta}}$

- $\beta^{\tau,\epsilon} = \frac{96\mathcal{I}(\mathcal{F}, \|\cdot\|_\infty)}{\lambda_{\mathrm{low}}^{\tau,\epsilon}} + 4\left(\frac{\|\mathcal{F}\|_\infty}{\lambda_{\mathrm{low}}^{\tau,\epsilon}} + m_c\right)\sqrt{2\log\frac{4}{\delta}},$

*then with probability at least* $1 - \delta$, *for all* $f \in \mathcal{F}$,

$$\widehat{R}_\rho^{\tau,\epsilon}(f) \ge R_{\rho - \frac{\alpha^{\tau,\epsilon}}{\sqrt{n}}}^{\tau,\epsilon}(f) \ge \mathbb{E}_{\zeta \sim Q}[f(\zeta)] - \epsilon D_{\mathrm{KL}}(\pi^{P,Q}\|\pi_0)$$

*whenever* $W_c^\tau(P, Q) \le \rho$.

*In particular, if* $m_c < \frac{\rho_{\mathrm{crit}}^{\tau,\epsilon}}{4}$ *and* $n > \frac{16(\alpha^{\tau,\epsilon} + \beta^{\tau,\epsilon})^2}{(\rho_{\mathrm{crit}}^{\tau,\epsilon} - 4m_c)^2}$, *then for any* $\rho > \max\left\{m_c, \frac{\alpha^{\tau,\epsilon}}{\sqrt{n}}\right\}$, *with probability at least* $1 - \delta$, *for all* $Q$ *such that* $W_c^\tau(P, Q) \le \rho$, $\widehat{R}_\rho^{\tau,\epsilon}(f) \ge \mathbb{E}_{\zeta \sim Q}[f(\zeta)] - \epsilon D_{\mathrm{KL}}(\pi^{P,Q}\|\pi_0)$ *for all* $f \in \mathcal{F}$.

*Proof.* Under Assumption 2.1, let $\lambda_{\mathrm{low}}^{\tau,\epsilon} > 0$ be given by Proposition E.2, and assume $\rho > m_c$. As for standard WDRO, our goal is to apply Proposition E.3 and to compute its constants thanks to Lemma C.3. By Lemma C.3, and taking $\lambda_{\mathrm{up}} = \frac{2\|\mathcal{F}\|_\infty}{\rho - m_c}$, we have the following constants:

- $\Phi = \|\mathcal{F}\|_\infty - (-\|\mathcal{F}\|_\infty - 2\lambda_{\mathrm{low}}^{\tau,\epsilon} m_c) = 2(\|\mathcal{F}\|_\infty + \lambda_{\mathrm{low}}^{\tau,\epsilon} m_c)$

- $\Psi = \frac{2\|\mathcal{F}\|_\infty}{\lambda_{\mathrm{low}}^{\tau,\epsilon}} + m_c$

- $\lambda_{\mathrm{up}} = \frac{2\|\mathcal{F}\|_\infty}{\rho - m_c}$ and $L_\psi = \|\mathcal{F}\|_\infty + \frac{1}{\lambda_{\mathrm{low}}^{\tau,\epsilon}} + \frac{2\|\mathcal{F}\|_\infty m_c \epsilon}{\epsilon(\rho - m_c) + 2\tau\|\mathcal{F}\|_\infty}$.

In Proposition E.3, $\alpha^{\tau,\epsilon}$ corresponds to $B(\delta)$ and $\beta^{\tau,\epsilon}$ corresponds to $C(\delta)$ with the quantities above. In this case, we easily verify that $\alpha^{\tau,\epsilon}$ and $\beta^{\tau,\epsilon}$ have the desired expressions.

By strong duality, Proposition B.2, and by the dual upper bound, Lemma D.3, $R_\varrho^{\tau,\epsilon}(f)$ and $\widehat{R}_\varrho^{\tau,\epsilon}(f)$ admit the representations

$$R_\varrho^{\tau,\epsilon}(f) = \inf_{\lambda \in [0, \lambda_{\mathrm{up}})}\left\{\lambda\varrho + \mathbb{E}_{\xi \sim P}[\phi^{\tau,\epsilon}(\lambda, f, \xi)]\right\}$$

$$\widehat{R}_\varrho^{\tau,\epsilon}(f) = \inf_{\lambda \in [0, \lambda_{\mathrm{up}})}\left\{\lambda\varrho + \mathbb{E}_{\xi \sim \widehat{P}_n}[\phi^{\tau,\epsilon}(\lambda, f, \xi)]\right\},$$

for any $\varrho \ge 0$ and $f \in \mathcal{F}$. Recall that $\rho > m_c$. If furthermore $\frac{\alpha^{\tau,\epsilon}}{\sqrt{n}} < \rho \le \frac{\rho_{\mathrm{crit}}^{\tau,\epsilon}}{4} - \frac{\beta^{\tau,\epsilon}}{\sqrt{n}}$, then with probability at least $1 - \delta$, we have for all $f \in \mathcal{F}$, $\widehat{R}_\rho^{\tau,\epsilon}(f) \ge R_{\rho - \frac{\alpha}{\sqrt{n}}}^{\tau,\epsilon}(f)$ by Proposition E.3 hence we obtain the first inequality.

Now, toward the second inequality, let $Q \in \mathcal{P}(\Xi)$ such that $W_c^\tau(P, Q) \leq \rho$. Let $\pi^{P,Q} \in \mathcal{P}(\Xi \times \Xi)$ satisfying $[\pi^{P,Q}]_1 = P$, $[\pi^{P,Q}]_2 = Q$ and $\mathbb{E}_{(\xi,\zeta) \sim \pi^{P,Q}}[c(\xi, \zeta)] + \tau D_{\mathrm{KL}}(\pi^{P,Q} \| \pi_0) = W_c^\tau(P, Q)$. We finally obtain for all $f \in \mathcal{F}$, $R_{\rho - \frac{\alpha}{\sqrt{n}}}^{\tau,\epsilon}(f) \geq \mathbb{E}_{\zeta \sim Q}[f(\zeta)] - \epsilon D_{\mathrm{KL}}(\pi^{P,Q} \| \pi_0)$.

As to the last statement, if $m_c < \frac{\rho_{\mathrm{crit}}^{\tau,\epsilon}}{4}$ and $n > \frac{16(\alpha^{\tau,\epsilon} + \beta^{\tau,\epsilon})^2}{(\rho_{\mathrm{crit}}^{\tau,\epsilon} - 4m_c)^2}$, then $\max\left\{ m_c, \frac{\alpha^{\tau,\epsilon}}{\sqrt{n}} \right\} < \frac{\rho_{\mathrm{crit}}^{\tau,\epsilon}}{4} - \frac{\alpha^{\tau,\epsilon}}{\sqrt{n}}$. For any $\max\left\{ m_c, \frac{\alpha^{\tau,\epsilon}}{\sqrt{n}} \right\} < \rho < \frac{\rho_{\mathrm{crit}}^{\tau,\epsilon}}{4} - \frac{\beta^{\tau,\epsilon}}{\sqrt{n}}$, the bound holds by the first part of the result. For $\rho \geq \frac{\rho_{\mathrm{crit}}^{\tau,\epsilon}}{4} - \frac{\beta^{\tau,\epsilon}}{\sqrt{n}}$, since $\widehat{R}_\rho^{\tau,\epsilon}(f)$ is non-decreasing with respect to $\rho$, the bound also holds. $\qquad\square$

### E.3 Excess risk bounds

In this part, we prove the excess risk bounds (Proposition 3.1 and Proposition 3.3). The proofs consist in adapting the previous proofs of the generalization bounds. For standard WDRO, the general excess bound specializes in the case of Wasserstein-$p$ costs and Lipschitz losses.

**Theorem E.3** (Excess risk WDRO). *Let $\alpha$ be given by Theorem E.1. Under Assumption 2.1, if $\rho \leq \frac{\rho_{\mathrm{crit}}}{4} - \frac{\alpha}{\sqrt{n}}$, then with probability at least $1 - \delta$, for all $f \in \mathcal{F}$,*

$$\widehat{R}_\rho(f) \leq R_{\rho + \frac{\alpha}{\sqrt{n}}}(f).$$

*In particular, if $c = d(\cdot, \cdot)^p$, where $p \geq 1$, and there exists $\mathrm{Lip}_{\mathcal{F}} > 0$ such that every $f \in \mathcal{F}$ is $\mathrm{Lip}_{\mathcal{F}}$-Lipschitz with respect to $d$, then*

$$\widehat{R}_\rho(f) \leq \mathbb{E}_{\xi \sim P}[f(\xi)] + \mathrm{Lip}_{\mathcal{F}}\left(\rho + \frac{\alpha}{\sqrt{n}}\right)^{\frac{1}{p}}.$$

*Proof.* By definition of $\lambda_{\mathrm{low}}$ and Proposition E.1 we can write for any $0 < \rho' \leq \frac{\rho_{\mathrm{crit}}}{4}$,

$$R_{\rho'}(f) = \inf_{\lambda \in [\lambda_{\mathrm{low}}, \lambda_{\mathrm{up}})} \{\lambda \rho' + \mathbb{E}_{\xi \sim P}[\phi(\lambda, f, \xi)]\}$$

leading to

$$R_{\rho'}(f) = \inf_{\lambda \in [\lambda_{\mathrm{low}}, \lambda_{\mathrm{up}})} \{\lambda \rho' + \mathbb{E}_{\xi \sim P}[\phi(\lambda, f, \xi)]\}$$

$$\geq \inf_{\lambda \in [\lambda_{\mathrm{low}}, \lambda_{\mathrm{up}})} \left\{\lambda \rho' + \mathbb{E}_{\xi \sim \widehat{P}_n}[\phi(\lambda, f, \xi)] - \lambda \frac{\mathbb{E}_{\xi \sim \widehat{P}_n}[\phi(\lambda, f, \xi)] - \mathbb{E}_{\xi \sim P}[\phi(\lambda, f, \xi)]}{\lambda}\right\}$$

$$\geq \inf_{\lambda \in [\lambda_{\mathrm{low}}, \lambda_{\mathrm{up}})} \left\{\lambda \rho' + \mathbb{E}_{\xi \sim \widehat{P}_n}[\phi(\lambda, f, \xi)] - \lambda \alpha_n\right\}$$

$$\geq \inf_{\lambda \in [\lambda_{\mathrm{low}}, \lambda_{\mathrm{up}})} \left\{\lambda \left(\rho' - \frac{B(\delta)}{\sqrt{n}}\right) + \mathbb{E}_{\xi \sim \widehat{P}_n}[\phi(\lambda, f, \xi)]\right\}$$

$$\geq \inf_{\lambda \in [0, \infty)} \left\{\lambda \left(\rho' - \frac{B(\delta)}{\sqrt{n}}\right) + \mathbb{E}_{\xi \sim \widehat{P}_n}[\phi(\lambda, f, \xi)]\right\} = \widehat{R}_{\rho' - \frac{B(\delta)}{\sqrt{n}}}(f).$$

whenever $\rho' > B(\delta)/\sqrt{n}$, and the inequality holds with probability at least $1 - \delta$ for all $f \in \mathcal{F}$. Recall also that $B(\delta) = \alpha$ (see the proof of Theorem E.1). Taking $\rho' = \rho + \frac{\alpha}{\sqrt{n}}$ leads to the desired result as long as $\rho + \frac{\alpha}{\sqrt{n}} \leq \frac{\rho_{\mathrm{crit}}}{4}$.

Toward a proof of the last part, assume that any $f \in \mathcal{F}$ is Lipschitz with constant $\mathrm{Lip}_{\mathcal{F}}$, and $c = d^p$ with $p \geq 1$. For any couple $(\xi, \zeta) \in \Xi \times \Xi$, we have

$$f(\zeta) \leq f(\xi) + \mathrm{Lip}_{\mathcal{F}} d(\xi, \zeta).$$

Integrating over an arbitrary coupling $\pi$ with first marginal $P$ and second marginal $Q$ satisfying $W_c(Q, P) \leq \rho + \alpha/\sqrt{n}$ gives

$$\mathbb{E}_Q[f] \leq \mathbb{E}_P[f] + \mathrm{Lip}_{\mathcal{F}} \mathbb{E}_\pi[d] \leq \mathbb{E}_P[f] + \mathrm{Lip}_{\mathcal{F}} \mathbb{E}_\pi[c]^{\frac{1}{p}}$$

where we used Jensen inequality. For any $Q$ satisfying $W_c(Q, P) \leq \rho + \alpha/\sqrt{n}$, taking the infimum in the above inequality over such couplings $\pi$, gives

$$\mathbb{E}_Q[f] \leq \mathbb{E}_P[f] + \mathrm{Lip}_{\mathcal{F}}(\rho + \alpha/\sqrt{n})^{\frac{1}{p}}$$

hence we obtain the result by definition of $R_{\rho + \frac{\alpha}{\sqrt{n}}}(f)$. $\qquad\square$

**Theorem E.4** (Excess risk for regularized WDRO). *Let $\alpha^{\tau,\epsilon}$ be given by Theorem E.2. Under Assumption 2.1, if $m_c < \rho \leq \frac{\rho^{\tau,\epsilon}_{\mathrm{crit}}}{4} - \frac{\alpha^{\tau,\epsilon}}{\sqrt{n}}$, then with probability at least $1 - \delta$, for all $f \in \mathcal{F}$,*

$$\widehat{R}^{\tau,\epsilon}_\rho(f) \leq R^{\tau,\epsilon}_{\rho+\frac{\alpha^{\tau,\epsilon}}{\sqrt{n}}}(f).$$

*Proof.* For $\rho > m_c$, strong duality holds (Proposition B.2). The proof is then identical to that of the standard WDRO setting (Theorem E.3). ☐

### E.4 GENERALIZATION CONSTANTS OF LINEAR MODELS

The two following results correspond to Proposition 3.2, which is the estimation of the constants $\rho_{\mathrm{crit}}$ and $\lambda_{\mathrm{low}}$ for linear models in the framework of Shafieezadeh-Abadeh et al. (2019). We assume the support of $P$ to belong to an Euclidean ball of diameter $D$ centered at zero. We then define $\Xi$ as the closed ball of diameter $3D$ centered at zero.

**Proposition E.4** (Linear regression). *Consider the parametric loss $f(\theta,(x,y)) = (\langle\theta,x\rangle - y)^2$, where $\theta$ belongs to a compact subset $\Theta \subset \mathbb{R}^p$, and the transport cost $c = \|\cdot - \cdot\|^2$. Assume there exists $\omega > 0$ such that*

$$\inf_{\theta\in\Theta} \|(\theta,-1)\|^2 \geq \omega.$$

*Then Theorem 3.1 and Proposition 3.1 hold with $\rho_{\mathrm{crit}} \geq D^2$ and $\lambda_{\mathrm{low}} \geq \frac{\omega}{2}$.*

*Proof.* In this setting, the expression of $\rho_{\max}$ is

$$\rho_{\max}(\lambda) = \inf_{\theta\in\Theta} \mathbb{E}_{\xi\sim P} \left[ \min \left\{ \|\xi - \zeta'\|^2 \; : \; \zeta' \in \arg\max_{\zeta\in\Xi}\{f(w,\zeta) - \lambda\|\xi - \zeta\|^2\} \right\} \right]$$

For any $\xi \in \Xi$ and $\theta \in \Theta$, the term inside the argmax writes

$$f(\theta,\zeta) - \lambda\|\xi - \zeta\|^2 = \zeta^T(M - \lambda I)\zeta + 2\lambda\zeta^T\xi - \lambda\|\xi\|^2.$$

Consider $\zeta = (u,v)$, $\xi = (u_0,v_0)$, $u,u_0 \in \mathbb{R}$, the representations in an orthonormal basis of $\mathbb{R}^p$, such that the first element $(\theta,-1)/\|(\theta,-1)\|$ is the eigen vector of $M$. We can write the above equation with $u$ and $v$ terms:

$$f(\theta,\zeta) - \lambda\|\xi - \zeta\|^2 = (\|(\theta,-1)\|^2 - \lambda)u^2 + 2\lambda u \cdot u_0 - \lambda\|v\|^2 + 2\lambda\langle v,v_0\rangle \qquad (34)$$

If $\lambda \leq \omega$ then we have $\|(\theta,-1)\|^2 - \lambda > 0$, hence the maximum of $f(\theta,\zeta) - \lambda\|\xi - \zeta\|^2$ with respect to $\zeta$ is only attained at the boundary of $\Xi$ (otherwise we could increase the quadratic term with respect to $u$). For all $\lambda \leq \omega$, we thus can bound from below

$$\rho_{\max}(\lambda) \geq \mathbb{E}_{\xi\sim P}[\min\|\xi - \zeta\|^2 \; : \; \|\zeta\| = 3D/2] \geq D^2.$$

In particular, $\rho_{\mathrm{crit}} \geq D^2$. We also remark that $\rho_{\mathrm{crit}} \leq 4D^2$ hence we have $2\lambda_{\mathrm{low}} \geq \omega$ by definition of $\lambda_{\mathrm{low}}$ (Proposition E.1). ☐

**Proposition E.5** (Logistic regression). *Consider the parametric loss $f(\theta,(x,y)) = \log(1 + e^{-y\langle\theta,x\rangle})$ where $\theta$ belongs to a compact subset $\Theta \subset \mathbb{R}^p$, and the transport cost $c = \|\cdot - \cdot\|^2$. Assume there exists $\omega > 0$ such that*

$$\inf_{\theta\in\Theta} \|\theta\|^2 \geq \omega.$$

*Then Theorem 3.1 and Proposition 3.1 hold with $\rho_{\mathrm{crit}} \geq D^2$ and $\lambda_{\mathrm{low}} \geq \frac{\omega}{8\left(1+e^{D\Omega}\right)}$, where $\Omega = \sup_{\theta\in\Theta} \|\theta\|^2$.*

*Proof.* For the logistic regression, we have

$$f(\theta, \zeta) - \lambda\|\zeta - \xi\|^2 = \log\left(1 + e^{\langle\theta,\zeta\rangle}\right) - \lambda\|\zeta - \xi\|^2. \tag{35}$$

Consider the representation $\zeta = s\theta + v$, where $s \in \mathbb{R}$ and $v$ is orthogonal to $\theta$. Then we have

$$f(\theta, \zeta) - \lambda\|\zeta - \xi\|^2 = \log\left(1 + e^{s\|\theta\|^2}\right) - s^2\lambda\|\theta\|^2 + 2s\lambda\langle\theta, \xi\rangle - \lambda\|v - \xi\|^2.$$

The second order derivative with respect to $s$ is

$$\frac{\|\theta\|^4}{\left(1 + e^{s\|\theta\|^2}\right)\left(1 + e^{-s\|\theta\|^2}\right)} - 2\lambda\|\theta\|^2. \tag{36}$$

The term $\left(1 + e^{s\|\theta\|^2}\right)\left(1 + e^{-s\|\theta\|^2}\right)$ is lower than $2\left(1 + e^{|s|\|\theta\|^2}\right) < 2\left(1 + e^{D\Omega}\right)$. Hence we easily deduce that (36) is positive for all $\zeta \in \Xi$ if $\lambda < \frac{\omega}{4\left(1 + e^{D\Omega}\right)}$. If this condition holds, then maximizers of $f(\theta, \zeta) - \lambda\|\zeta - \xi\|^2$ for $\zeta \in \Xi$ are included in the boundary of $\Xi$, meaning that $\rho_{\max}(\lambda) \geq D^2$ if $\lambda \leq \frac{\Omega}{4\left(1 + e^{D\Omega}\right)}$. Since $\rho_{\mathrm{crit}} \leq 4D^2$, then $2\lambda_{\mathrm{low}} \geq \frac{\omega}{4\left(1 + e^{D\Omega}\right)}$. $\qquad\square$

# F  SIDE REMARKS

This part contains results supporting various remarks made in the main text.

## F.1  INTERPRETATION OF THE CRITICAL RADIUS

The results of this part justify the interpretation of the radius made in Remark 3.1 and Proposition F.3

**Proposition F.1.** *If $\rho \geq \rho_{\mathrm{crit}}$, then there exists $f \in \mathcal{F}$ such that*

$$R_\rho(f) = \max_{\xi \in \Xi} f(\xi).$$

*In particular, in the setting of Theorem 3.1, if $\rho \geq \rho_{\mathrm{crit}} + \frac{\alpha}{\sqrt{n}}$, with probability at least $1 - \delta$, there exists $f \in \mathcal{F}$ such that*

$$\widehat{R}_\rho(f) = \max_{\xi \in \Xi} f(\xi).$$

*Proof.* The first part is identical to the square cost case, see (Azizian et al., 2023a, Remark 3.2 ). The second part is obtained by Theorem E.1: in the setting of Theorem E.1, we have with probability at least $1 - \delta$, $\widehat{R}_\rho(f) \geq R_{\rho-\alpha/\sqrt{n}}$ for all $f \in \mathcal{F}$. Hence if $\rho \geq \rho_{\mathrm{crit}} + \alpha/\sqrt{n}$, we obtain the result by applying the first part to the radius $\rho - \alpha/\sqrt{n}$. $\qquad\square$

The following result gives an interpretation of the critical radius $\rho_{\mathrm{crit}}^{\tau,\epsilon}$ in regularized WDRO appearing in Theorem 3.2. We show that when the radius $\rho$ is larger than this value, then some robust losses become degenerated. Precisely, they become independent of $\rho$ and are equal to a regularized version of the worst-case loss $\max_\Xi f$.

**Proposition F.2.** *Assume $\rho \geq \rho_{\mathrm{crit}}^{\tau,\epsilon}$. Then there exists $f \in \mathcal{F}$ such that*

$$R_\rho^{\tau,\epsilon}(f) = \sup_{\substack{\pi \in \mathcal{P}(\Xi \times \Xi) \\ [\pi]_1 = P}} \left\{ \mathbb{E}_{\zeta \sim [\pi]_2}[f(\zeta)] - \epsilon D_{\mathrm{KL}}(\pi \| \pi_0) \right\}.$$

*In particular, in the setting of Theorem 3.2, if $\rho \geq \rho_{\mathrm{crit}}^{\tau,\epsilon} + \frac{\alpha^{\tau,\epsilon}}{\sqrt{n}}$, with probability at least $1 - \delta$, there exists $f \in \mathcal{F}$ such that*

$$\widehat{R}_\rho^{\tau,\epsilon}(f) = \sup_{\substack{\pi \in \mathcal{P}(\Xi \times \Xi) \\ [\pi]_1 = P}} \left\{ \mathbb{E}_{\zeta \sim [\pi]_2}[f(\zeta)] - \epsilon D_{\mathrm{KL}}(\pi \| \pi_0) \right\}.$$

*Proof.* In the regularized case, we can verify that the critical radius has the expression

$$\rho_{\text{crit}}^{\tau,\epsilon} = \inf_{f \in \mathcal{F}} \left\{ \mathbb{E}_{\xi \sim P} \left[ \mathbb{E}_{\zeta \sim \pi_0^{f/\epsilon}(\cdot|\xi)} \left[ \frac{\tau}{\epsilon} f(\zeta) + c(\xi,\zeta) \right] - \tau \log \mathbb{E}_{\zeta \sim \pi_0(\cdot|\xi)} e^{\frac{f(\zeta)}{\epsilon}} \right] \right\}, \quad (37)$$

see for instance the proof of Lemma D.2. Let $f \in \mathcal{F}$ be arbitrary. Consider a coupling $\pi^* \in \mathcal{P}(\Xi \times \Xi)$ such that $[\pi^*]_1 = P$ and $\pi^*(\cdot|\xi) = \pi_0^{\frac{f}{\epsilon}}(\cdot|\xi)$ for almost all $\xi \in \Xi$. We first verify that for a good choice of $f$, it is included in the uncertainty set defining $R_\rho^{\tau,\epsilon}(f)$.

We compute $D_{\text{KL}}(\pi^* \| \pi_0)$. Below, we set $Z(\xi) := \mathbb{E}_{\zeta \sim \pi_0(\cdot|\xi)} \left[ e^{\frac{f(\zeta)}{\epsilon}} \right]$.

$$\begin{aligned} D_{\text{KL}}(\pi^* \| \pi_0) &= \mathbb{E}_{\xi \sim P} \left[ \mathbb{E}_{\zeta \sim \pi_0^{\frac{f}{\epsilon}}(\cdot|\xi)} \left[ \log \left( \frac{e^{\frac{f(\zeta)}{\epsilon}}}{Z(\xi)} \right) \right] \right] \\ &= \mathbb{E}_{\xi \sim P} \left[ \mathbb{E}_{\zeta \sim \pi_0^{\frac{f}{\epsilon}}(\cdot|\xi)} \left[ \frac{f(\zeta)}{\epsilon} \right] - \log \mathbb{E}_{\zeta \sim \pi_0(\cdot|\xi)} \left[ e^{\frac{f(\zeta)}{\epsilon}} \right] \right] \\ &= \mathbb{E}_{(\xi,\zeta) \sim \pi^*} \left[ \frac{f(\zeta)}{\epsilon} \right] - \mathbb{E}_{\xi \sim P} \left[ \log \mathbb{E}_{\zeta \sim \pi_0(\cdot|\xi)} \left[ e^{\frac{f(\zeta)}{\epsilon}} \right] \right]. \end{aligned} \quad (38)$$

This leads to

$$\mathbb{E}_{\pi^*}[c] + \tau D_{\text{KL}}(\pi^* \| \pi_0) = \mathbb{E}_{\xi \sim P} \left[ \mathbb{E}_{\zeta \sim \pi_0^{f/\epsilon}(\cdot|\xi)} \left[ \frac{\tau}{\epsilon} f(\zeta) + c(\xi,\zeta) \right] - \tau \log \mathbb{E}_{\zeta \sim \pi_0(\cdot|\xi)} e^{\frac{f(\zeta)}{\epsilon}} \right]$$

which is the term in the infimum (37). Since $f$ was chosen arbitrary, this means that if $\rho > \rho_{\text{crit}}^{\tau,\epsilon}$, then there exists $f \in \mathcal{F}$ such that the coupling $\pi^*$ defined above (depending on $f$) satisfies $\mathbb{E}_{(\xi,\zeta) \sim \pi^*}[c(\xi,\zeta)] + \tau D_{\text{KL}}(\pi^* \| \pi_0) \leq \rho$, and we obtain

$$R_\rho^{\tau,\epsilon}(f) \geq \mathbb{E}_{\zeta \sim [\pi^*]_2}[f(\zeta)] - \epsilon D_{\text{KL}}(\pi^* \| \pi_0).$$

On the other hand by the computation (38), we have

$$R_\rho^{\tau,\epsilon}(f) \geq \mathbb{E}_{\zeta \sim [\pi^*]_2}[f(\zeta)] - \epsilon D_{\text{KL}}(\pi^* \| \pi_0) = \epsilon \mathbb{E}_{\xi \sim P} \left[ \log \mathbb{E}_{\zeta \sim \pi_0(\cdot|\xi)} \left[ e^{\frac{f(\zeta)}{\epsilon}} \right] \right]. \quad (39)$$

By Donsker-Varadhan variational formula Donsker & Varadhan (1975), for almost all $\xi \in \Xi$, we have

$$\log \mathbb{E}_{\zeta \sim \pi_0(\cdot|\xi)} \left[ e^{\frac{f(\zeta)}{\epsilon}} \right] = \sup_{\nu \in \mathcal{P}(\Xi)} \left\{ \mathbb{E}_{\zeta \sim \nu}[f(\zeta)/\epsilon] - D_{\text{KL}}(\nu \| \pi_0(\cdot|\xi)) \right\}. \quad (40)$$

Reinjecting (40) in (39) gives

$$\begin{aligned} R_\rho^{\tau,\epsilon}(f) &\geq \epsilon \mathbb{E}_{\xi \sim P} \left[ \sup_{\nu \in \mathcal{P}(\Xi)} \left\{ \mathbb{E}_{\zeta \sim \nu}[f(\zeta)/\epsilon] - D_{\text{KL}}(\nu \| \pi_0(\cdot|\xi)) \right\} \right] \\ &\geq \sup_{\substack{\pi \in \mathcal{P}(\Xi \times \Xi) \\ [\pi]_1 = P}} \left\{ \mathbb{E}_{\xi \sim P} \left[ \mathbb{E}_{\zeta \sim \pi(\cdot|\xi)}[f(\zeta)] - \epsilon D_{\text{KL}}(\pi(\cdot|\xi) \| \pi_0(\cdot|\xi)) \right] \right\} \\ &= \sup_{\substack{\pi \in \mathcal{P}(\Xi \times \Xi) \\ [\pi]_1 = P}} \left\{ \mathbb{E}_{\zeta \sim [\pi]_2}[f(\zeta)] - \epsilon D_{\text{KL}}(\pi \| \pi_0) \right\}, \end{aligned}$$

where we used the chain rule for $D_{\text{KL}}$ divergence (see e.g. Theorem 2.15 in Polyanskiy & Wu. (2023)): $D_{\text{KL}}(\pi \| \pi_0) = \mathbb{E}_{\xi \sim P}[D_{\text{KL}}(\pi(\cdot|\xi) \| \pi_0(\cdot|\xi))] + D_{\text{KL}}([\pi]_1 \| [\pi_0]_1) \geq \mathbb{E}_{\xi \sim P}[D_{\text{KL}}(\pi(\cdot|\xi) \| \pi_0(\cdot|\xi))]$. Since we clearly have $R_\rho^{\tau,\epsilon}(f) \leq \sup_{\substack{\pi \in \mathcal{P}(\Xi \times \Xi) \\ [\pi]_1 = P}} \left\{ \mathbb{E}_{\zeta \sim [\pi]_2}[f(\zeta)] - \epsilon D_{\text{KL}}(\pi \| \pi_0) \right\}$, this yields the first part.

The second part is a direct consequence of the generalization bound Theorem E.2 as for the standard case (see the proof of Proposition F.1). □

The following result allows to interpret $\rho_{\text{crit}}$ as a fluctuations measure on $\mathcal{F}$:

**Proposition F.3** (Critical radius and loss fluctuations). *Suppose* $\text{supp } P = \Xi$. *Under Assumption 2.1* $\rho_{\text{crit}} > 0$ *if and only if* $\mathcal{F}$ *contains no constant functions.*

*Proof.* We prove the equivalence by contraposition. For the first implication, assume there exists $f \in \mathcal{F}$ such that for all $\xi$ and $\zeta$ in $\Xi$, $f(\xi) = f(\zeta)$. This means for all $\xi \in \Xi$ we have $\xi \in \arg\max_\Xi f$, whence $\min\{c(\xi, \zeta) : \zeta \in \arg\max_\Xi f\} = 0$ because $c(\xi, \xi) = 0$. By integrating with respect to $\xi \sim P$, we obtain $\rho_{\text{crit}} = 0$.

Now, towards the other implication we assume $\rho_{\text{crit}} = 0$ and we show that $\mathcal{F}$ contains a constant function. Let $(f_k)_{k \in \mathbb{N}}$ be a sequence of functions from $\mathcal{F}$ such that

$$\mathbb{E}_{\xi \sim P}\left[\min\left\{c(\xi, \zeta) : \zeta \in \arg\max_\Xi f_k\right\}\right] \xrightarrow[k \to \infty]{} 0.$$

By compactness of $\mathcal{F}$, Assumption 2.1.3, we may assume $(f_k)_{k \in \mathbb{N}}$ to converge to some function $f^*$ for the norm $\|\cdot\|_\infty$. The set-valued map $f \mapsto \arg\max_\Xi f$ is outer semicontinuous with compact values (Lemma A.2) and $c$ is jointly continuous, hence for any $\xi \in \Xi$, the function $f \mapsto \min\{c(\xi, \zeta) : \zeta \in \arg\max_\Xi f\}$ is lower semicontinuous on $\mathcal{F}$, see Lemma A.1. Thus we have

$$\liminf_{k \to \infty} \min\left\{c(\xi, \zeta) : \zeta \in \arg\max_\Xi f_k\right\} \geq \min\left\{c(\xi, \zeta) : \zeta \in \arg\max_\Xi f^*\right\}$$

by lower semicontinuity (see Definition A.1).

By integration with respect to $\xi \sim P$, we have

$$0 = \liminf_{k \to \infty} \mathbb{E}_{\xi \sim P}\left[\min\left\{c(\xi, \zeta) : \zeta \in \arg\max_\Xi f_k\right\}\right]$$

$$\geq \mathbb{E}_{\xi \sim P}\left[\liminf_{k \to \infty} \min\left\{c(\xi, \zeta) : \zeta \in \arg\max_\Xi f_k\right\}\right]$$

$$\geq \mathbb{E}_{\xi \sim P}\left[\min\left\{c(\xi, \zeta) : \zeta \in \arg\max_\Xi f^*\right\}\right]$$

hence $\mathbb{E}_{\xi \sim P}[\min\{c(\xi, \zeta) : \zeta \in \arg\max_\Xi f^*\}] = 0$. Finally, for $P$-almost all $\xi \in \Xi$, $\min\{c(\xi, \zeta) : \zeta \in \arg\max_\Xi f^*\} = 0$. This means by compactness of $\arg\max_\Xi f^*$ that $\xi \in \arg\max_\Xi f^*$ for $P$-almost all $\xi \in \Xi$. Since $\text{supp} P = \Xi$ and $f^*$ is continuous, we obtain that $f^*$ is constant. $\square$

## F.2 NECESSITY OF THE DUAL UPPER BOUND

We exhibit an example where the function $\mu \mapsto \psi^{\tau,\epsilon}(\mu, f, \xi)$ is not Lipschitz as $\mu \to 0$. This justifies the necessity of bounding the dual solution above in the regularized case, as done in Lemma D.3.

**Proposition F.4.** *Consider $\tau = 0$, $\epsilon > 0$, $\Xi = [0, 1]$, $c(\xi, \zeta) = |\xi - \zeta|$ and assume that the reference distribution is a truncated Laplace $\pi_0(\mathrm{d}\zeta|\xi) \propto e^{-|\xi - \zeta|}\mathbb{1}_{[0,1]}(\zeta)\mathrm{d}\zeta$. Assume furthermore $\mathcal{F}$ is a family of functions from $[0, 1]$ to $\mathbb{R}$ which satisfies $e^{-\frac{2\|\mathcal{F}\|_\infty}{\epsilon}} \geq \epsilon$.*

*Then for almost all $\xi \in [0, 1]$ and all $f \in \mathcal{F}$, $\mu \mapsto \psi^{\tau,\epsilon}(\mu, f, \xi)$ is not Lipschitz at $0^+$.*

*Proof.* Let $\xi \in (0, 1)$ and $f \in \mathcal{F}$. The expression of the derivative of $\psi^{0,\epsilon}$ with respect to $\mu$ is given by (16):

$$\partial_\mu \psi^{0,\epsilon}(\mu, f, \xi) = \mathbb{E}_{\zeta \sim \pi_0^{\frac{\mu f - c(\xi, \cdot)}{\mu\epsilon}}(\cdot|\xi)}\left[\frac{\epsilon c(\xi, \zeta)}{\mu\epsilon}\right] + \epsilon \log \mathbb{E}_{\zeta \sim \pi_0(\cdot|\xi)}\left[e^{\frac{\mu f(\zeta) - c(\xi, \zeta)}{\mu\epsilon}}\right].$$

In particular, it satisfies

$$\partial_\mu \psi^{0,\epsilon}(\mu, f, \xi) \leq e^{\frac{2\|\mathcal{F}\|_\infty}{\epsilon}} \mathbb{E}_{\zeta \sim \pi_0^{-\frac{c(\xi, \cdot)}{\mu\epsilon}}(\cdot|\xi)}\left[\frac{c(\xi, \zeta)}{\mu}\right] + \epsilon \log \mathbb{E}_{\zeta \sim \pi_0(\cdot|\xi)}\left[e^{-\frac{c(\xi, \zeta)}{\mu\epsilon}}\right] + \|\mathcal{F}\|_\infty. \quad (41)$$

On the other hand, by Donsker-Varadhan formula Donsker & Varadhan (1975), we can write

$$\log \mathbb{E}_{\zeta \sim \pi_0(\cdot|\xi)}\left[e^{-\frac{c(\xi, \zeta)}{\mu\epsilon}}\right] = \mathbb{E}_{\zeta \sim \pi_0^{-\frac{c(\xi, \zeta)}{\mu\epsilon}}(\cdot|\xi)}\left[\frac{-c(\xi, \zeta)}{\mu\epsilon}\right] - D_{\text{KL}}\left(\pi_0^{-\frac{c(\xi, \cdot)}{\mu\epsilon}}(\cdot|\xi)\bigg\|\pi_0(\cdot|\xi)\right).$$

Reinjecting this in (41) and using $e^{-\frac{2\|\mathcal{F}\|_\infty}{\epsilon}} \geq \epsilon$ gives

$$\partial_\mu \psi^{\tau,\epsilon}(\mu, f, \xi) \leq \|\mathcal{F}\|_\infty - D_{\mathrm{KL}}\left(\pi_0^{-\frac{c(\xi,\cdot)}{\mu\epsilon}}(\cdot|\xi)\middle\|\pi_0(\cdot|\xi)\right).$$

Consequently, to prove non-Lipschitzness of $\psi^{0,\epsilon}(\cdot, f, \xi)$ at 0, we show that

$$D_{\mathrm{KL}}\left(\pi_0^{-\frac{c(\xi,\cdot)}{\mu\epsilon}}(\cdot|\xi)\middle\|\pi_0(\cdot|\xi)\right) \to \infty$$

as $\mu \to 0$. We show that $\pi_0^{-\frac{|\xi-\cdot|}{\mu\epsilon}}(\cdot|\xi)$ converges in law to $\delta_\xi$. Let $\varphi : \mathbb{R} \to \mathbb{R}$ be of class $C^\infty$ with compact support. With the change of variable $u \leftarrow \frac{\xi-\zeta}{\mu\epsilon}$, we have

$$\int_0^1 e^{-\frac{|\xi-\zeta|}{\mu\epsilon}}\varphi(\zeta)\mathrm{d}\zeta = \mu\epsilon\int_{\mathbb{R}} \mathbb{1}_{\left[\frac{\xi-1}{\mu\epsilon}, \frac{\xi}{\mu\epsilon}\right]}(u)e^{-|u|}\varphi(\xi + \mu\epsilon u)\mathrm{d}u.$$

Also, we easily verify that

$$\int_0^1 e^{-\frac{|\xi-\zeta|}{\mu\epsilon}}\mathrm{d}\zeta = \int_0^\xi e^{-\frac{\xi-\zeta}{\mu\epsilon}}\mathrm{d}\zeta + \int_\xi^1 e^{-\frac{\zeta-\xi}{\mu\epsilon}}\mathrm{d}\zeta = \mu\epsilon(2 - e^{-\frac{\xi}{\mu\epsilon}} - e^{\frac{-(1-\xi)}{\mu\epsilon}}),$$

hence we obtain

$$\mathbb{E}_{\zeta \sim \pi_0^{-\frac{|\xi-\cdot|}{\mu\epsilon}}}[\varphi(\zeta)] = \frac{\int_{\mathbb{R}} \mathbb{1}_{\left[\frac{\xi-1}{\mu\epsilon}, \frac{\xi}{\mu\epsilon}\right]}(u)e^{-|u|}\varphi(\xi + \mu\epsilon u)\mathrm{d}u}{2 - e^{-\frac{\xi}{\mu\epsilon}} - e^{\frac{-(1-\xi)}{\mu\epsilon}}}. \tag{42}$$

We then have the following:

- $2 - e^{-\frac{\xi}{\mu\epsilon}} - e^{\frac{-(1-\xi)}{\mu\epsilon}}$ converges to 2 as $\mu \to 0$,

- For all $u \in \mathbb{R}$, $\mathbb{1}_{\left[\frac{\xi-1}{\mu\epsilon}, \frac{\xi}{\mu\epsilon}\right]}(u)e^{-|u|}\varphi(\xi + \mu\epsilon u)\mathrm{d}u$ converges to $e^{-|u|}\varphi(\xi)$ as $\mu \to 0$, hence its integral with respect to $u$ converges to $2\varphi(\xi)$ by dominated convergence theorem.

Combining both limits in (42) gives $\mathbb{E}_{\zeta \sim \pi_0^{-\frac{|\xi-\cdot|}{\mu\epsilon}}(\cdot|\xi)}[\varphi(\zeta)] \to \varphi(\xi)$. This means that $\pi_0^{-\frac{|\xi-\cdot|}{\mu\epsilon}}(\cdot|\xi)$ converges in law to $\delta_\xi$. We have $D_{\mathrm{KL}}(\delta_\xi\|\pi_0(\cdot|\xi)) = \infty$, hence by lower semicontinuity of the $D_{\mathrm{KL}}$-divergence for the convergence in law (or weak convergence), see e.g. Theorem 4.9 from Polyanskiy & Wu. (2023), we get $D_{\mathrm{KL}}\left(\pi_0^{-\frac{c(\xi,\cdot)}{\mu\epsilon}}(\cdot|\xi)\middle\|\pi_0(\cdot|\xi)\right) \xrightarrow[\mu\to 0]{} \infty$. This means that $\psi^{0,\epsilon}(\cdot, f, \xi)$ is not Lipschitz near 0. $\square$

### F.3 ON CONTINUITY OF MAXIMIZERS

We justify the importance of relaxing Assumption 5.1 from Azizian et al. (2023a) which corresponds to compactness of $\mathcal{F}$ with respect to the distance $D_{\mathcal{F}}(f,g) := \|f - g\|_\infty + d_H(\arg\max_\Xi f, \arg\max_\Xi g)$. We show that this condition is actually equivalent to assuming continuity on $f \mapsto \arg\max f$, which is a strong condition and difficult to verify in practice.

**Proposition F.5.** *For $(f,g) \in \mathcal{F} \times \mathcal{F}$, define*

$$D_{\mathcal{F}}(f,g) := \|f - g\|_\infty + d_H(\arg\max_\Xi f, \arg\max_\Xi g)$$

*where $d_H$ is the Hausdorff distance on the set of compact subsets of $\Xi$, $\mathcal{K}(\Xi)$. Assume $(\mathcal{F}, \|\cdot\|_\infty)$ is compact. Then we have the equivalence*

$$(\mathcal{F}, D_{\mathcal{F}}) \text{ is compact} \iff f \mapsto \arg\max_\Xi f \text{ is continuous from } (\mathcal{F}, \|\cdot\|_\infty) \text{ to } (\mathcal{K}(\Xi), d_H).$$

*Proof.* We prove ($\Rightarrow$). Assume $(\mathcal{F}, D_{\mathcal{F}})$ is compact. Let $f \in \mathcal{F}$, and let $(g_k)_{k \in \mathbb{N}}$ be an arbitrary sequence from $\mathcal{F}$ such that $g_k$ converges to $f$ for $\| \cdot \|_\infty$. We want to show that $\arg\max_{\sqsubseteq} g_k$ converges to $\arg\max_{\sqsubseteq} f$ for $d_H$, proving the continuity of the arg max map. By compactness of $(\mathcal{F}, D_{\mathcal{F}})$, $(g_k)_{k \in \mathbb{N}}$ admits accumulation points for $D_{\mathcal{F}}$. Let $h$ be any one of them. We may extract a subsequence from $(g_k)_{k \in \mathbb{N}}$ converging to $h$, say $g_{n_k} \underset{k \to \infty}{\to} h \in \mathcal{F}$. In particular, $g_{n_k}$ converges to $h$ for $\| \cdot \|_\infty$. We necessarily have $h = f$ by definition of the sequence $(g_k)_{k \in \mathbb{N}}$. It means that $(g_k)_{k \in \mathbb{N}}$ admits only one possible accumulation point for $D_{\mathcal{F}}$, which is $f$. This implies $g_k$ converges to $f$ for $D_{\mathcal{F}}$, hence $\arg\max_{\sqsubseteq} g_k$ converges to $\arg\max_{\sqsubseteq} f$.

Now, we prove ($\Leftarrow$). Let $(f_k)_{k \in \mathbb{N}}$ be a sequence from $\mathcal{F}$. By compactness of $(\mathcal{F}, \| \cdot \|_\infty)$, we may extract a converging subsequence $f_{n_k} \underset{k \to \infty}{\to} f$ for $\| \cdot \|_\infty$. Assuming $f \mapsto \arg\max_{\sqsubseteq} f$ is continuous gives that $\arg\max_{\sqsubseteq} f_{n_k}$ converges to $\arg\max_{\sqsubseteq} f$, which is the desired result. $\qquad\square$

