# OpenReview forum: "Universal generalization guarantees for Wasserstein distributionally robust models"
_ICLR.cc/2025/Conference — ICLR 2025 Spotlight_

### Official Review · Reviewer_hWjP · 2024-11-02

**Soundness:** 3
**Presentation:** 4
**Contribution:** 3
**Rating:** 8
**Confidence:** 4

**Summary:**

The proposed paper provides lower bounds on the robust empirical risk under unorthodox but interesting scaling limits on the radius of the Wasserstein ball around the empirical risk.  The paper uses some cool techniques which are not often seen in machine learning.

The paper is relatively clearly written.  However, I think there are a few little things here and there which are either difficult to justify (in the current form) or perhaps not well-defined (see below). Also, the introduction is excessively general while the setting rapidly collapses to a much more specific setting shortly after.

**Strengths:**

The paper is well written, interesting, and theoretical and provides very nice lower bounds on the robust empirical risk.  The results are nice, and so is the use of set-valued analysis to derive them.  Several relevant examples are considered, making a large portion of how these results can be used nearly transparent.

**Weaknesses:**

Nevertheless,  I think some of the assumptions are a bit opaque (see below), and I'm not certain some quantities are well-defined.

*Minor*
- Citing Cuturi and Perée's book is odd when mentioning the Wasserstein distance.  Perhaps the original source or a book on optimal transport such as Villani's book, would be more natural, IMO.

- The definition of Wasserstein distance circa (1) is incorrect, $\Xi$ must be Polish, and  c *must* be a power $p\in [1,\infty)$ of a *metric* topologizing $\Xi$; what you write is just some transport problem.  Eg if c is not symmetric, then $W_c$ is not a metric in general, or if $c(x,y)=0$ for all $x,y$ then $W_c$ cannot separate points.

- Perhaps "suitable" distributions is more appropriate before (1), since the distance explodes if these have no finite moment.

- Line 65: bad grammar: "it does not introduce approximate term" also imprecise.

- Line 66: Is Wainright's book and Boucheron the best reference?  Perhaps older papers, e.g. on VC dimension, Bartlett's old papers on Rademacher complexity, or old papers on chaining are more natural references?

- Line 69: "This theoretical feature is specific to WDRO and highlights its potential to give more resilient models." This can be **much** less hand-wavy.  Please explain more precisely/mathematically.

- Line 149: In a metric space $(x,..)$ not "In (X,..) a metric space".

- Assumption 1 vs. Line 145: You say that $\Xi$ is just a measurable space, then later you say its a compact metric space.  Why no   be forthright and say its a metric space on line 145.  Similarly, why is $\mathcal{F}$ an arbitrary family of functions, then straightaway after is actually a compact set of continuous functions.

- Line 176: Why jointly Lipschitz?  If $\Theta$ is compact, then since you already assumed $\Xi$ is compact, then it is enough for $\Theta\times\Xi\ni (\theta,\xi)\mapsto f(\theta,\xi)\in \mathbb{R}$ to be continuous; to deduce the compactness of $\{f(\theta,\cdot):\,\theta \in \Theta\}$ by the currying Lemma.

- Line 176: Not sure why you say "if $\Xi$ is compact, since this was assumed a few lines earlier on the same page.

- Maybe more natural examples come from Arzela-Ascoli...

- Should the definition of the Dudley entropy integral really be in a footnote, while more basic ideas are in the main text.

- Line 221: The words "the metric" are missing.

- Line 223: There are many more references of the use of this type of metric, especially in exponential convergence rate results for Markov chains (wrt $W_1$ over countable metric spaces with this distance).

- Line 245: "sample randomness" (I know what you mean...but the word independent is misleading as this

- Assumption 1: Why call (2) jointly continuous, it is just standard continuity (actually inform continuity by compactness).

**Questions:**

- Why not submit to JMLR?  The paper is very rigorous and rather long and technical for an ML conference?  You also examine the problem in good detail.

- Could you provide a simple example in Theorem 3.2, where the optimal coupling is known under (say) Gaussianity assumptions?

- I'm a bit confused.  What does $\operatorname{argmax}_{\Xi}\,f$ mean in (5) a sup norm or something?

- Why is $\min\{ c(\xi,\zeta): ... \}$ measurable?  In particular, (independent of the meaning of the argmax, above question), why is there a measurable selection $\xi\mapsto \zeta$?  Without this, its not clear that $\rho_{\operatorname{crit}}$ is well defined.  I'm guessing this is Berge's theorem (which is in Aliprantis & Border) somehow, but please spell it out for us :)

- Each result assumes that the (difficult to interpret) $\rho_{\operatorname{crit}}$ is "large enough".  Can you please provide a general set of conditions ensuring that $\rho_{\operatorname{crit}}$ can be bounded away from $0$.

- Is it fair to compare, verbally, our results to those of Fournier et al. (and similar bounds, say, found in [1])?  Since you are considering a small ball around the empirical measure while their results guarantee a minimal radius such that the empirical measure contains the true measure whp.   Furthermore, those rates are only tight (afaik) when the measure is very spread out; more precisely, it is Alhors $d$-regular, see e.g. [3] for a nice clean proof.

- In theorem 3.2, why is $\pi^{P,Q}\ll \pi_0$?  To be this isn't directly evident... I.e.\ why is the RHS not trivially $-\infty$ in general?



[1] Graf, Siegfried, and Harald Luschgy. Foundations of quantization for probability distributions. Springer Science & Business Media, 2000.
[2] Otto, Felix, and Cédric Villani. "Generalization of an inequality by Talagrand and links with the logarithmic Sobolev inequality." Journal of Functional Analysis 173.2 (2000): 361-400.
[3] Kloeckner, Benoit. "Approximation by finitely supported measures." ESAIM: Control, Optimisation and Calculus of Variations 18.2 (2012): 343-359.

---

> ### Author Response · Authors · 2024-11-21
> **Response to main questions**
>
> Thank you for your reading and positive feedback. Your several remarks helped us improving the paper and we addressed each of them in our revision. Please find below our answers to the important points from your review.
>
>
> ### Questions
>
> > Why not submit to JMLR? The paper is very rigorous and rather long and technical for an ML conference? You also examine the problem in good detail.
>
> Yes we tried to propose a thorough study. We acknowledge that certain parts of our work are technical. We have made an effort to structure them rigorously and have placed them in the appendices. In the main, we have instead focused on the context, the results, and the key ideas that underpin their derivation. We believe that this material will be valuable to a significant portion of the community, whether they work on theoretical and practical aspects.
>
>
> > Could you provide a simple example in Theorem 3.2, where the optimal coupling is known under (say) Gaussianity assumptions?
>
> This is not evident even for simple examples: the optimal coupling depends on $Q$ which can be any distribution satisfying $W_c^\tau(P,Q) \leq \rho$. In order to gain more interpretable results, approximation results might be established to quantify how much the right hand side is close to the exact risk. Considering existing works on this aspect, see e.g. [1], we believe this would require more structure on $c$. This is a relevant and non trivial question.
>
> > I'm a bit confused. What does $\operatorname{argmax}_\Xi f$ mean in (5) a sup norm or something?
>
> This is the set of maximizers of $f$ on $\Xi$, the points attaining the maximum of $f$ on $\Xi$, the ``argmax''. This now appear in the notations.
>
> > Why is $\operatorname{min} c(\xi,\zeta) ...$ measurable?
>
> As you say, this can be given by the measurable Maximum Theorem. In our proofs, we did not discuss measurability of every function because it was often satisfied through stronger notions such as continuity or semicontinuity (for instance to prove Lemma 4.1 in appendix D.1).
>
>
> > Each result assumes that the (difficult to interpret)  $\rho\_{\text{crit}}$ is "large enough" Can you please provide a general set of conditions ensuring that $\rho\_{\text{crit}}$ can be bounded away from 0.
>
> This is quite abstract, indeed. We interpret it as follows: when $P$ is supported on $\Xi$, $\rho_{\text{crit}} > 0$ if and only if $\mathcal{F}$ contains no constant function. We have a proposition in appendix establishing this; we have added a pointer to it in the main, for more clarity.
>
>
> > In theorem 3.2, why is $\pi^{P,Q} \ll \pi_0$?
>
> This is due to the definition of KL divergence. This is now mentioned as a footnote.
>
>
>
>
> > Is it fair to compare, verbally, our results to those of Fournier et al [...] ?
>
>
> Fournier and Guillin concentration result can indeed be improved when some structure can be leveraged. In this paper, we consider instead the general case, with no restriction on specific distributions; this is the setting of the seminal work of Mohajerin Esfahani and Kuhn (2018), relying on Fournier and Guillin, for the statistical results.  In our work, our take is that, in this general setting, we can still gain in sample complexity, not using the structure of the distributions, but rather by leveraging the structure of the WDRO optimization problem itself.
>
>
>
> [1] Waiss Azizian, Franck Iutzeler, and Jerome Malick. Regularization for wasserstein distributionally robust optimization. ESAIM: Control, Optimisation and Calculus of Variations, 29:33, 2023b.

---

> ### Author Response · Authors · 2024-11-21
> **Response to minor points**
>
> We now continue with your minor points. We have addressed all of them in the revision. Please find our answer to the most important ones below.
>
>
> ### Minor points
> > The definition of Wasserstein distance circa (1) is incorrect,
> $\Xi$ must be Polish, and $c$ must be a power $p \geq 1$
>  of a metric topologizing $\Xi$
> ; what you write is just some transport problem [...]
>
> **On the terminology ''Wasserstein distance'' (1):** Your are right, writing "Wasserstein distance" in our case is a slight abuse of terminology. WDRO may be studied for general costs (see e.g. [2]), under our Assumption 2.1. Note that we also require $c(\xi,\zeta)$ to be zero if and only if $\xi = \zeta$.  As you say, this setting does not imply $W_c$ to be a metric. We now use the terminology "optimal transport cost" as in [2] to avoid any confusion.
>
>
> > **Assumption 1 vs. Line 145** You say that is just a measurable space, then later you say its a compact metric space [...]
>
>
> The section notation was indeed a bit general for the main text. We modified it to make it more simple and we wrote a general notation part in the appendix for the proofs. In the main text we now define the spaces $\Xi$, $\mathcal{F}$ and the cost $c$ right from the beginning in the notation part. We kept their specificities (such as compactness or continuity) in Assumption 2.1 to facilitate the comparison of our setting with the literature, and make the presentation as much transparent as  possible.
>
>
> > Line 69: "This theoretical feature is specific to WDRO and highlights its potential to give more resilient models." This can be much less hand-wavy. Please explain more precisely/mathematically.
>
>
> This relates to the previous sentence, which already describe the inequality (3). We reorganized both sentences to make it clearer
>
>
>
>
>
> > Line 176: Why jointly Lipschitz? [...]
>
> This is a good remark, there was an omission here. Joint Lipschitz continuity ensures $\mathcal{I}\_{\mathcal{F}}$ is finite; this is now added in the revision.  $\mathcal{I}\_{\mathcal{F}} < \infty$ can also be satisfied when assuming $f(\cdot,\xi)$ $L$-Lipschitz for all $\xi \in \Xi$. For simplicity we assume joint Lipschitz continuity since we can not think of a relevant situation where $\Xi$, $\Theta$ would be compact but $f$ not jointly Lipschitz continuous.
>
>
> > Line 223: There are many more references of the use of this type of metric
>
> This is indeed a natural metric, which exists in many other contexts than WDRO.
>
> > Assumption 1: Why call (2) jointly continuous [...]
>
> Yes, but this is continuity on the product space $\Xi \times \Xi$.  We use the terminology joint continuity to avoid any confusion with continuity with respect to each variable.
>
>
>
> [2] Jose Blanchet and Karthyek Murthy. Quantifying distributional model risk via optimal transport.
> Mathematics of Operations Research, 44(2):565–600, 2019.

---

### Official Review · Reviewer_tsmX · 2024-11-04

**Soundness:** 3
**Presentation:** 3
**Contribution:** 3
**Rating:** 6
**Confidence:** 3

**Summary:**

This paper provides exact generalization guarantees for Wasserstein Distributionally Robust Optimization (WDRO) for a wide variety of models with compactness and finite Dudley's entropy assumptions. The results apply to radius $\rho$ scaling as $O(1/\sqrt{n})$, which does not suffer from the curse of dimensionality. The results also cover the double regularization case.

**Strengths:**

- The generalization guarantees of this work do not rely on restrictive assumptions like smoothness compared to the previous work (Azizian et al. 2023a).
- This paper is well-structured, and the theoretical results and proof sketches are clearly presented.

**Weaknesses:**

- In Section 3.2, the authors discussed how their results on generalization guarantees apply to linear regression and logistic regression. However, more complicated models such as neural networks with ReLU or other smooth activation functions (e.g. GELU) are not discussed.
- The theoretical results require a lower bound on $n$, while Theorem 3.4 of Azizian et al. (2023a) applies to all $n \ge 1$. The implications of this requirement should be discussed.

**Questions:**

- What are the practical implications of the generalization guarantees compared to Azizian et al. (2023a)? Can you provide some numerical results analogous to Appendix H of Azizian et al. (2023a)?

---

> ### Author Response · Authors · 2024-11-21
>
> Thank you for your positive feedback. Please find our response to your several questions below.
>
>
> > In Section 3.2, the authors discussed how their results on generalization guarantees apply to linear regression and logistic regression. However, more complicated models such as neural networks with ReLU or other smooth activation functions (e.g. GELU) are not discussed.}
>
>
> We underline that more complicated models, e.g. neural networks with ReLU, are covered by our general analysis, in contrast to the previous ones.
>
>  This being said, we agree that obtaining theoretical or empirical insights into the constants $\lambda_{\text{low}}$ and $\rho_{\text{crit}}$ in deep learning contexts would be interesting. This question remains open and deserves a dedicated study.
>
> ---
> > The theoretical results require a lower bound on $n$, while Theorem 3.4 of Azizian et al. (2023a) applies to all
> $n \geq 1$. The implications of this requirement should be discussed.
>
>
> This is a good technical point to raise, thank you. Theorems 3.1 and 3.4 of Azizian et al. (2023a) also (implicitly) require such lower bounds. Let us explain why.
>
> Theorems 3.1 and 3.4 from Azizian et al. 2023 require to choose $\rho$ as
> $$\frac{\alpha}{\sqrt{n}}\leq \rho  \leq \frac{\rho_{\text{crit}}}{2} - \frac{\beta}{\sqrt{n}}$$
> (here we replaced their big O notation by some constants $\alpha >0$ and $\beta > 0$). However, this range can be empty with no further condition on $n$. Only for $n$ high enough $\rho$ can be chosen in the interval $[\frac{\alpha}{\sqrt{n}}, \frac{\rho_{\text{crit}}}{2} - \frac{\beta}{\sqrt{n}}]$. This gives the condition $\frac{\alpha}{\sqrt{n}} \leq  \frac{\rho_{\text{crit}}}{2} - \frac{\beta}{\sqrt{n}}$ and then the lower bound $n \geq 4 (\alpha + \beta)^2/\rho_{\text{crit}}^2$. In this case, as we did, we may also remove the upper bound on $\rho$ by monotonicity of the robust loss with respect to $\rho$. We added a comments on this point in appendix.
>
>
> ---
> > What are the practical implications of the generalization guarantees compared to Azizian et al. (2023a)? Can you provide some numerical results analogous to Appendix H of Azizian et al. (2023a)?
>
>
> It would indeed be great to have numerical illustrations of generalization properties in the general setting we consider. This is however a difficult point that would require a complete and rigorous work. This is out of scope of this theoretical paper. Among the many challenges, efficient practical procedures to compute WDRO models in deep learning are still missing. This is how we conclude the paper insisting on future work.

---

### Official Review · Reviewer_tSiX · 2024-11-04

**Soundness:** 3
**Presentation:** 3
**Contribution:** 3
**Rating:** 8
**Confidence:** 4

**Summary:**

This paper presents novel bounds on for the DRO loss using the Wasserstein distance. In particular, they address the question of finding the minimal $\rho$ used by the empirical robust loss such that the loss is an upper bound for the actual population loss. The main challenge is to overcome the dependency on the distance between W(P_n, P) \sim n^{1/d}. While this problem has been studied in the literature, and dimension free bounds exist, this paper presents a proof requiring weaker assumptions.




---------- after the rebuttal ------------

I thank the authors for their response and increased the score accordingly.

**Strengths:**

The paper addresses an important problem is generalization bounds/theoretic ML. In particular:

- The paper is well written and the results are nicely presented.
- The proof sketch in Section 4 is excellent. It is very easy to follow and often neglected in these types of papers
- The proof idea is smart, non-trivial and interesting.

**Weaknesses:**

Given that this is a more traditional field, I would expect a clearer comparison with the existing works. While the authors do a very good job in presenting the proof idea, it is not so clear how the proof fundamentally differs from existing works.

**Questions:**

I am happy to increase my score and support this paper with a high confidence if the authors can provide an extensive discussion during the rebuttal on the assumptions in Azizian et al. (2023a) . In particular, my two major questions are: can the authors be more precise in which cases their assumptions are weaker than the ones in  Azizian et al. (2023a). In particular, can you give an example for a class of distributions that are covered by this paper but not by  Azizian et al. (2023a)? Moreover, can the authors explain why the proof in  Azizian et al. (2023a) breaks for your assumptions and why it is not trivial to extend the proof?



Smaller question:
Isn't assumption 3.1 (1) always true satisfied by w<=1. Is it possible that this is a typo?

**Details Of Ethics Concerns:**

-

---

> ### Author Response · Authors · 2024-11-21
>
> Thank you very much for your positive feedback, all your comments/remarks, and your two main questions above. It helps us improving our presentation, and especially on the position w.r.t Azizian et al. 2023a, which is the closest work.
>
> In the revision, we now provide a discussion, just below our main assumptions (Assumption 2.1) where we compare our setting to Azizian et al. 2023a in details. This complements the key differences, already mentioned in the related work section (lines 120 to 130).
>
>
> **Let us recall here our improvements, compared to Azizian et al (2023a):**
> - The setting of Azizian et al (2023a) restricts to smooth functions $f \in \mathcal{F}$ (twice differentiable with  uniformly bounded derivatives on a convex sample space). We only require the $f \in \mathcal{F}$ to be continuous on a metric space. In addition to nonsmooth functions, this allows us to consider distributions on sample spaces with discrete and continuous variables (as for e.g. classification tasks).
>
>
>
> - Their proof require to take $c$ as the squared norm and the
> reference distribution  $\pi_0(\cdot|\xi)$ as a Gaussian distribution. We consider instead general costs $c$, continuous with respect to a distance on $\Xi$ and an arbitrary reference probability distribution.
>
>
>
> For instance, this setting is captured by us but not by Azizian et al. (2023a):
>
> > (i) The sample space $\Xi =  B(0,R) \times \{0,1\}$ where $R > 0$
> >
> > (ii) The loss family $\mathcal{F} = \\{ f(\theta, \cdot) \ : \ \theta \in \Theta \\}$ with the cross entropy loss $$f(\theta, x,y) = - y \log(h(\theta,x)) - (1 - y) \log(h(\theta,x))$$
>  where  $h(\theta, \cdot)$ is a feedforward network with RELU activations and $\Theta$ is compact.
> >
> > (iii) The cost function:  $c((x,y), (x',y')) = \|x-x'\|_{p}^{q} + \kappa \mathbb{1}\_{y \neq y'}$
>
>
>
>
> - Moreover, in their proof, to overcome nonsmoothness of WDRO  (which poses concerns for applying concentration results), they require two technical assumptions: a compactness condition (1) and growth conditions around maximizers (2), this is their **Assumption 5**:
>
>
> (1) For any $R > 0$, there exists $\Delta > 0$ such that
>    $$\forall f \in \mathcal{F}, \; \forall \zeta \in \Xi, \; d\left(\zeta, \operatorname{argmax} f\right) \geq R \implies f(\zeta) - \max f \leq -\Delta.$$
>
>
> (2) There exist $\mu > 0$ and $L > 0$ such that, for all $f \in \mathcal{F}$, $\xi \in \Xi$ and $\xi^*$ a projection of $\xi$ on $\operatorname{argmax} f$,
>     $$f(\xi^*) \geq f(\xi) + \frac{\mu}{2} \|\xi - \xi^*\|^2 - \frac{L}{6} \|\xi - \xi^*\|^3.$$
>
> We do not rely on these conditions. They are rather strong and difficult to verify since maximizers over the sample space are hard to control in general and both depend on the sample space and the function class geometries. In particular, (1) requires $\mathcal{F}$ to be compact with respect to a distance defined by summing the sup norm and the Hausdorff distance between maximizers sets. Equivalently, this can be seen as the continuity of $f \mapsto \operatorname{argmax} f$ on $\mathcal{F}$ (see our Proposition F.4 in appendix), which is hard to verify.
>
>
>
> The main technical difficulties in our proof was thus to get rid Assumption 5 from Azizian et al 2023a and to deal with the nonsmooth aspect of the WDRO objective. To this purpose, we simplified the proof and use nice nonsmooth analysis tools. We highlight this aspect in the sketch of proof, section 4.2 ''Definition of the lower bound" where we present a maximal radius function.
>
>
>
> **About your smaller question:** Indeed, the assumption is vacuous in this case and we may take $\omega = 1$. We decided to keep the constant $\omega$ in the results in order to highlight the dependence of $\lambda_{\text{low}}$ on the hypothesis domain. To make our statements more precise, we added the condition $\omega \geq 1$ in Assumption 3.1.1.

---

### Meta-Review · Area_Chair_VJTG · 2024-12-18

**Metareview:**

This paper analyzes distributionally robust optimization, which effectively addresses data uncertainty and distribution shifts in training robust machine learning models.   Existing generalization guarantees are often approximate or limited to specific cases with hard-to-verify assumptions.  This paper establishes exact generalization guarantees that apply to a broad range of scenarios, including deep learning models with nonsmooth activations, and provides an excess bound on the robust objective along with an extension to Wasserstein robust models with entropic regularizations.

The paper considers an important problem, and is generally well written (including the proofs and theorem statements).  The rebuttal helped a lot in clarifying the advantage of the new result over the existing ones.  Please polish the revised PDF and make sure that the discussions from the rebuttal and discussion period are properly incorporated.

**Additional Comments On Reviewer Discussion:**

The rebuttal has been noted by the reviewers and have been taken into account by the AC in the recommendation of acceptance/rejection.

---

### Decision · Program_Chairs · 2025-01-22

Accept (Spotlight)